# Robust Concept Erasure via Kernelized Rate-Distortion Maximization

**Somnath Basu Roy Chowdhury**
UNC Chapel Hill

**Nicholas Monath**
Google DeepMind

**Avinava Dubey**
Google Research

**Amr Ahmed**
Google Research

**Snigdha Chaturvedi**
UNC Chapel Hill
{somnath, snigdha}@cs.unc.edu
{nmonath, avinavadubey, amra}@google.com

## Abstract

Distributed representations provide a vector space that captures meaningful relationships between data instances. The distributed nature of these representations, however, entangles together multiple attributes or *concepts* of data instances (e.g., the topic or sentiment of a text, characteristics of the author (age, gender, etc), etc). Recent work has proposed the task of *concept erasure* [50, 52], in which rather than making a concept predictable, the goal is to remove an attribute from distributed representations while retaining other information from the original representation space as much as possible. In this paper, we propose a new distance metric learning-based objective, the **K**ernelized **Ra**te-Distortion **M**aximizer (KRaM), for performing concept erasure. KRaM fits a transformation of representations to match a specified distance measure (defined by a labeled concept to erase) using a modified rate-distortion function. Specifically, KRaM's objective function aims to make instances with similar concept labels dissimilar in the learned representation space while retaining other information. We find that optimizing KRaM effectively erases various types of concepts—categorical, continuous, and vector-valued variables—from data representations across diverse domains. We also provide a theoretical analysis of several properties of KRaM's objective. To assess the quality of the learned representations, we propose an alignment score to evaluate their similarity with the original representation space. Additionally, we conduct experiments to showcase KRaM's efficacy in various settings, from erasing binary gender variables in word embeddings to vector-valued variables in GPT-3 representations.

## 1 Introduction

Learned representations, particularly distributed representations [30], are at the core of machine learning with applications in natural language [39], images [21], biology [4], physics [3], and several other domains [40, 48]. These vector-based representations of data instances create an inner product space where similarities and nearest-neighbor relationships are "meaningful". However, due to the distributed nature of these representations, the definition of "meaningful" is often not easily discernible. In other words, shared properties of data instances nearby in the vector space are not always evident. These shared properties are often referred to as *concepts* [31]. For instance, concepts in representations of images include objects in the image, whether it is indoor or outdoor, etc. Similarly, concepts in text representations include the topics, and characteristics of the author (e.g., geographic location, gender, etc.). For applications that necessitate conditioning on specific attributes to make or explain predictions, these distributed representations can pose challenges. Consequently,

a significant body of work has focused on jointly learning representations and disentangling their underlying concepts [28, 29].

However, state-of-the-art representations for tasks across various domains often come from pre-trained models (e.g., ViTs [23] for images, GPT [14] for text, among others). In many cases, these pre-trained representations are utilized directly without fine-tuning the original model, due to factors such as computational burden or limited model API access [24]. This presents a challenge for fitting disentangled representations of data instances and their concepts since the model parameters are frozen. Rather, it presents an opportunity for learning a transformation (or similarly learning a distance metric) for the pre-trained representations.

While learning such a transformation of pre-trained representations can be used for many applications (e.g., classification, regression, etc.) that involve disentangling specific concepts from representations, we focus on the recently proposed task of *concept erasure* [50]. The objective of concept erasure is twofold: (1) to learn representations that minimize the classification accuracy (or mean squared error) for a specific concept variable and (2) to retain as much other information from the original representations as possible. This becomes possible when the representations are not correlated with the concept variable. We should note that there are indeed some trivial methods to reduce the correlation – such as generating random representations or making all representations identical. However, these solutions fail to retain any information from the original representation space. This highlights one of the key challenges in concept erasure – retaining information from the original space while removing a given concept. This challenge is accentuated by the lack of a *pre-specified* downstream task for which the representations could be optimized. Instead, objectives for concept erasure use no supervision (apart from the labels of the concept to erase). This makes it difficult to use adversarial learning [60] or mutual information estimation [17, 43] methods for concept erasure.

The independence of concept erasure from down-stream tasks makes it amenable to a variety of applications that use the modified representations. For example, when developing a toxicity classifier for online comments, an organization may seek to ensure that content from different religious backgrounds is treated equitably. This can be achieved using concept erasure to remove information about religion from the text representations [60]. Concept erasure has also shown promise in interpreting the decision-making of large models by studying counterfactual scenarios where certain properties of the input are erased from intermediate layers [26].

In this paper, we present a new distance metric learning-based objective for concept erasure. We refer to our objective, **K**ernelized **Ra**te-Distortion **M**aximizer (KRaM). The objective fits a transformation of representations to match a specified distance measure using a kernelized rate-distortion function, where the kernel is constructed using concept labels. Specifically, KRaM's objective function tries to make instances with similar concept labels dissimilar in the learned representation space (Figure 1). Empirically, we find that optimizing KRaM results in representations that are uncorrelated with the concept variable, effectively leading to its erasure. To evaluate the quality of the representations, we propose a $k$-nearest neighbour based measure to capture the alignment of the learned representations with the original representation space. We conduct extensive experiments to demonstrate that KRaM is capable of erasing various types of concepts—categorical, continuous, and vector-valued variables—from data representations across a wide range of domains. We also theoretically analyze several properties of the proposed objective function. Our primary contributions are:

- We introduce a novel framework – KRaM, which uses a kernelized formulation of the rate-distortion function that is able to delete a range of concepts – categorical, continuous, or vector-valued variables from representations (Section 3).
- We propose a computationally efficient alignment measure, to evaluate how informative the learned representations are about the original representation space (Section 4).
- We perform a theoretical analysis of KRaM's objective and alignment measure. We conduct extensive experiments to showcase the efficacy of KRaM in a range of settings, from erasing binary gender variables in word embeddings to vector-valued variables in GPT-3 representations (Section 5).

## 2 Preliminaries & Background

In this section, we first formally describe the concept erasure setup, then discuss some prior concept erasure techniques, and finally introduce the fundamentals of rate-distortion theory.

**Problem Setup**. In *concept erasure* [50, 52], we consider the input representations $x \in \mathcal{X}$ and the concept $a \in \mathcal{A}$ as random variables. We assume access to samples $[(x_1, a_1), (x_2, a_2), \ldots]$ drawn from the joint distribution $P(\mathcal{X}, \mathcal{A})$. The goal of concept erasure is to learn a function $f(\cdot)$ that generates representations $[f(x_1), f(x_2), \ldots] \in \mathcal{Z}$, such that it is infeasible to predict the concept labels $a \in \mathcal{A}$ from $\mathcal{Z}$. In addition to erasing the concept variable, $f(x) \in \mathcal{Z}$ should retain as much information about $x \in \mathcal{X}$ as possible. We do not impose any constraints on the nature of the concept. It can be: categorical ($a \in [k]$), continuous ($a \in \mathbb{R}$), or vector-valued ($a \in \mathbb{R}^d$) random variable.

Concept Erasure is also closely related to learning invariant representations with respect to an attribute through adversarial learning [8, 25, 32, 60]. However, concept erasure differs from adversarial learning in two key aspects: (a) the input representations $x \in \mathcal{X}$ remain frozen during the concept erasure process (only erasure function $f$ is updated), and (b) it does not rely on a specific downstream task. This setup is beneficial in situations where we can access representations but lack the necessary resources or infrastructure to train or fine-tune the model that generated them. In the following section, we describe the details of the proposed concept erasure framework, KRaM.

**Prior Work**. Concept erasure was initially introduced by [13] in the context of removing binary gender labels from GloVe embeddings [46]. Initial works on this problem [13, 49] performed concept erasure by projecting representations onto the null space of the optimal separating linear subspace for the categorical concept. Recent work [50] has introduced a generalized objective for this solution, presenting it as a minimax game between concept identification and nullspace projection, and further provided a closed-form solution for its relaxed convex version. Nonetheless, these techniques are limited by two main assumptions: (a) the erasure function $f(\cdot)$ is linear, and (b) the concept variable is categorical. A linear erasure function ensures that a linear subspace, which could identify the concept label for instances, does not exist in the learned representation space, $\mathcal{Z}$. Consequently, it prevents any linear network from extracting the concept labels from the learned representations. However, it can still be possible for a non-linear network to predict the concept labels by identifying a non-linear concept subspace. Given that most modern ML architectures rely on non-linear networks, it is crucial to ensure that the concept is inaccessible to non-linear networks. More recent works have made progress towards non-linear concept erasure. This has been done either by using a linear concept erasure after projecting the input into a non-linear feature space [52], or directly utilizing a non-linear erasure function $f$ using a rate-distortion objective [18]. Despite their potential, these concept erasure techniques require access to categorical concept labels for all instances. Hence, it is not possible to erase other forms of concept variables (continuous or vector-valued) using these techniques without discretization of the concept labels, which often leads to information loss. In contrast to these techniques, our erasure framework KRaM is able to handle a variety of concept variables (categorical, continuous, or vector-valued) while performing non-linear concept erasure. This becomes possible as KRaM presumes access to a kernel matrix that is defined by the concept labels, and does not impose any additional constraints on the nature of the concept variable. Next, we discuss the fundamentals of the rate-distortion function that forms a building block of our framework.

**Rate Distortion**. In information theory [20], the compactness of a distribution is measured by their *coding length* – the number of binary bits required to encode it. In lossy data compression, a set of vectors $\mathcal{Z} = \{z_1, \ldots, z_n\} \in \mathbb{R}^{n \times d}$, sampled from a distribution $P(\mathcal{Z})$, is encoded using a coding scheme, such that the transmitted vectors $\{\hat{z}_i\}_{i=1}^n$ can be recovered up to a distortion $\epsilon$. The minimal number of bits required per vector to encode the sequence $\mathcal{Z}$ is defined by the *rate-distortion* function $R(\mathcal{Z}, \epsilon)$. The optimal $R(\mathcal{Z}, \epsilon)$ for vectors $\mathcal{Z}$ sampled from a multivariate Gaussian $\mathcal{N}(0, \Sigma)$ is:

$$R(\mathcal{Z}, \epsilon) = \frac{1}{2} \log_2 \det \left( I + \frac{d}{n\epsilon^2} \mathcal{Z}\mathcal{Z}^T \right), \tag{1}$$

where $n$ is the number of vectors and $d$ is the dimension of individual vectors. Equation 1 provides a tight bound even in cases where the underlying distribution $P(\mathcal{Z})$ is degenerate [38]. The rate-distortion function is also closely tied to the sphere packing problem [38] and represents the volume (or intrinsic dimension) of a representation set. Recent works like MCR$^2$ [59] have built on the rate-distortion function to learn discriminative representations for classification tasks. Concept erasure techniques [18, 19], have also used the rate-distortion function to erase categorical variables. Even though KRaM uses the rate-distortion function similar to these techniques, it is more versatile and capable of erasing different types of concept variables. KRaM also imposes additional constraints on the feature space to ensure robust concept erasure, which we discuss in the following section.

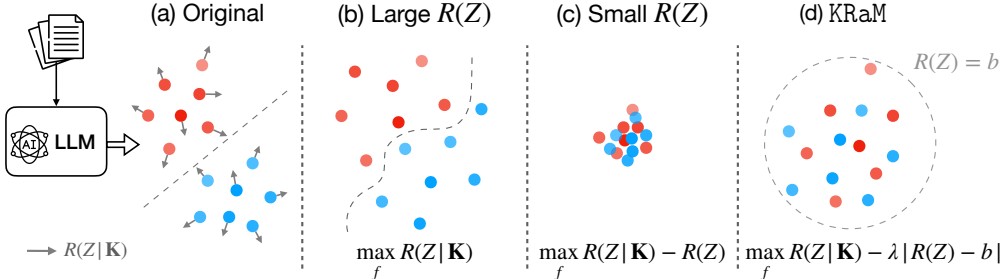

Figure 1: An illustration of concept erasure using KRaM. Input representations are retrieved from a large language model. The original representations (a) encodes a binary concept variable (the two classes are shown in ○ and ●), which we aim to erase. $R(\mathcal{Z}|\mathbf{K})$ term forces instances from the same class to move apart. However, for robust concept erasure the size of the representation space $R(Z)$ matters, which we illustrate visually in (b), (c), and (d). KRaM enforces the constraint $R(Z) = b$ to erase the concept while retaining information from original representations.

## 3   Kernelized Rate-Distortion Maximizer (KRaM)

We proceed by discussing how a concept variable can be erased from a representation set. A concept cannot be extracted by a predictive network (which is equivalent to erasure) if there is minimal correlation between the representation set, $\mathcal{Z}$, and the concepts, $\mathcal{A}$ [58]. Note that distances in the representation space can be indicative of the concept variable. For example, in Figure 1 (a) we observe that instances in the original representation space with similar concept labels (shown by their color ○ and ●) appear close to each other. Here, the distance between instances is correlated with the concept label, thereby making it feasible to identify the concept labels via a linear or non-linear boundary. Ideally, we want a representation space where distances are not reflective of concept labels, where instances with different concept labels appear together (e.g., Figure 1 (c) & (d)).

The intuition behind our approach, KRaM, is to make the distances in the learned representation space, $\mathcal{Z}$, uncorrelated with the concept variable, $\mathcal{A}$. Specifically, we try to make instance pairs similar in the concept space to be distant (or dissimilar) in the learned representation space, $\mathcal{Z}$. We achieve this by learning an erasure function $f$ (parameterized by a neural network) to transform the input $x \in \mathcal{X}$ into $\mathcal{Z}$. We propose a kernelized formulation of the rate-distortion function to train $f$:

$$R(\mathcal{Z}|\mathbf{K}) = \frac{1}{2} \log_2 \det \left( I + \frac{d}{n\epsilon^2} \mathcal{Z}\mathcal{Z}^T \odot \mathbf{K} \right), \tag{2}$$

where $\mathcal{Z} = f(\mathcal{X}) \in \mathbb{R}^{n \times d}$ and the kernel matrix, $\mathbf{K} \in \mathbb{R}^{n \times n}$, captures similarities between concept labels. The entries of the kernel matrix are inversely proportional to the distance between concept labels $\mathbf{K}_{ij} \propto 1/\mathrm{d}(a_i, a_j)$, where $\mathrm{d}(\cdot, \cdot)$ can be an arbitrary symmetric distance function, such that $\mathrm{d}(x, x) = 0$. We observe that $R(\mathcal{Z}|\mathbf{K})$ is sensitive to the scale of the representations $f(x) \in \mathcal{Z}$. Therefore, we fix the Frobenius norm of the representations using a layer normalization layer [5] to make $f(x) \in \mathbb{S}^d$, ensuring that individual instances have an equal impact on the loss.

Next, we discuss how maximizing $R(\mathcal{Z}|\mathbf{K})$ (Equation 2) helps in concept erasure. We proceed by noting that maximizing the standard version of the rate-distortion function (Equation 1) is equivalent to increasing the covariance of the representations, $\mathcal{Z}\mathcal{Z}^T$. In the kernelized rate-distortion function (Equation 2), we observe that the kernel matrix $\mathbf{K}$ assigns higher weights to instance pairs that have similar concept labels ($\mathbf{K}_{ij} \propto 1/\mathrm{d}(a_i, a_j)$). Intuitively, this means that maximizing $R(\mathcal{Z}|\mathbf{K})$ results in an increased dissimilarity between instance pairs with similar concept labels, as illustrated by the arrows in the center of Figure 1(a). This gradually leads to the distances in the learned representation space $\mathcal{Z}$ being unrelated to the concept labels.

However, simply maximizing $R(\mathcal{Z}|\mathbf{K})$ may not guarantee robust concept erasure without imposing additional constraints on the overall feature space. We explain why this happens by considering a few scenarios. First, consider the scenario where we do not impose any constraints on the feature space, shown in Figure 1(b). In this scenario, the volume (or intrinsic dimension) of the feature space (analogous to $R(\mathcal{Z})$ term) also expands as $R(\mathcal{Z}|\mathbf{K})$ is maximized (Lemma 1). Here, we observe that even though the intra-group distances have increased it is still possible to separate the two groups

using a non-linear decision boundary. Second, we consider the scenario where we try to minimize the volume of the feature space, which is equivalent to minimizing $R(\mathcal{Z})$. This is illustrated in Figure 1(c), where all instances are pushed together making it hard to predict the concept labels. However, it also results in a significant loss of information from the original representations (as the volume or intrinsic dimension collapses). As different instances become almost similar it destroys the unique features present in Figure 1(a), potentially rendering them ineffective for downstream tasks. Thus, it appears that the optimal approach is to maintain a constant size of the feature space as illustrated in Figure 1(d). We verify these scenarios empirically in Section 5.2. With this consideration, we present the following objective:

$$\max_f R(\mathcal{Z}|\mathbf{K}), \text{ subject to } R(\mathcal{Z}) = b, \tag{3}$$

where $b = R(\mathcal{X})$ is the initial number of bits required to encode the data and $\mathcal{Z} = f(\mathcal{X})$. In practice, we found that satisfying the equality $R(\mathcal{Z}) = b$ using a Lagrangian function hinders the maximization of $R(\mathcal{Z}|\mathbf{K})$. For concept erasure, we only want the feature space volume to be constant and do not require it to exactly be $b$. Therefore, we optimize a relaxed version of the objective (Equation 3):

$$\max_f R(\mathcal{Z}|\mathbf{K}) - \lambda|R(\mathcal{Z}) - b|, \tag{4}$$

where $\lambda$ is a hyperparameter. The second term in Equation 4 penalizes the network, $f$, if the overall volume $R(\mathcal{Z})$ deviates too far from $b$. Depending on the nature of the attribute (categorical, continuous, or vector-valued), the user can define the kernel matrix $\mathbf{K}$ between concept labels. For categorical concept variables, in our experiments, we use the kernel matrix whose values are $\mathbf{K}_{ij} \in \{0, 1\}$, where $\mathbf{K}_{ij} = 1$ if $a_i = a_j$ otherwise $\mathbf{K}_{ij} = 0$. For continuous and vector-valued variables, the kernel matrix can be derived from the concept labels by using a suitable kernel function (e.g., Gaussian, Laplacian, or Cauchy kernels) if it is not specified by the user. In our experiments, we use a Gaussian (RBF) kernel function for continuous and vector-valued concepts.

**Lemma 1** (General Bounds for $R(\mathcal{Z}|\mathbf{K})$). *For any set of representations $\mathcal{Z} \in \mathbb{R}^{n \times d}$, a kernel matrix $\mathbf{K} \in \mathbb{R}^{n \times n}$ using a kernel function satisfying $k(x, x) = 1$ and $\epsilon > 0$, it holds that:*

$$R(\mathcal{Z}) \le R(\mathcal{Z}|\mathbf{K}) \le \frac{n}{2} \log_2 \left(1 + d/n\epsilon^2\right), \tag{5}$$

*where the first equality is satisfied when $\mathbf{K} = \mathbf{1}\mathbf{1}^T$ and the second equality when $\mathcal{Z}\mathcal{Z}^T = I$.*

The detailed proof is provided in Appendix A.1. This result shows that $R(\mathcal{Z}|\mathbf{K})$ has a lower bound equal to the rate-distortion function of the representations with the upper bound being independent of the kernel matrix. We empirically show that maximizing $R(\mathcal{Z}|\mathbf{K})$ also results in an increase in $R(\mathcal{Z})$ in Section 5.2. Using the results of the above lemma, we can show that the proposed objective (Equation 4) is bounded in the following corollary.

**Corollary 1.** *Using assumptions in Lemma 1, for $\lambda \in [0, 1]$ the objective function (Equation 4) is bounded between $[-\lambda b, \max\{(1 + \lambda)U - \lambda b, (1 - \lambda)U + \lambda b\}]$, where $U = \frac{n}{2} \log_2 \left(1 + d/n\epsilon^2\right)$.*

## 4 Measuring Alignment

A limitation of prior works [18, 49, 50] is the lack of evaluation beyond proxy tasks of how information preserved by the learned representations $\mathcal{Z}$ about the original representations $\mathcal{X}$, which we refer to as *alignment*. An erasure framework that generates random representations is able to erase concept $\mathcal{A}$ perfectly but does not retain any information from $\mathcal{X}$. Therefore, it is important to measure the alignment as well while optimizing the erasure function $f$. To this end, we propose a computationally efficient measure by computing the overlap between $k$-nearest neighbour sets of $\mathcal{X}$ and $\mathcal{Z}$. The average nearest neighbour overlap across all representations is the alignment score ($A_k$) for a given concept erasure function $f$:

$$A_k(f) = \mathop{\mathbb{E}}_{x \sim \mathcal{X}} \left[|\text{knn}(x) \cap \text{knn}(f(x))|\right] / k, \tag{6}$$

where $\text{knn}(\cdot)$ function computes the $k$-nearest neighbour set of a representation. Alignment scores lie between $A_k(f) \in [k/n, 1]$ (Lemma 2). Notice that this measure is quite similar to non-parametric approaches for mutual information (MI) estimation [41, 35]. These methods, while leveraging nearest neighbour information, compute data statistics within a hypercube. However, these MI

estimates are often biased for high-dimensional data [27]. In contrast to these methods, we utilize the bijective mapping (erasure function $f$) between $\mathcal{X}$ and $\mathcal{Z}$ (which is not available in the general case of MI estimation) to compute the overlap between nearest neighbour sets. The computation of $A_k$ can be made faster using an efficient nearest neighbour data structure like $k$d-tree [10]. Note that a similar measure was used to determine the stability of word embeddings [57].

Empirically, we find that $A_k$ is well correlated with the downstream performance of $\mathcal{Z}$. Specifically, we perform a synthetic experiment to simulate concept erasure and assess the efficacy of $A_k$ in capturing the alignment of $\mathcal{Z}$, where we sample a representation set ($x \sim \mathbb{R}^{100}$) and their corresponding labels. We remove information from these representations by projecting them onto nullspaces of its dominant eigenvectors (details provided in Algorithm 1). During this process, we measure the prediction accuracy for the original labels

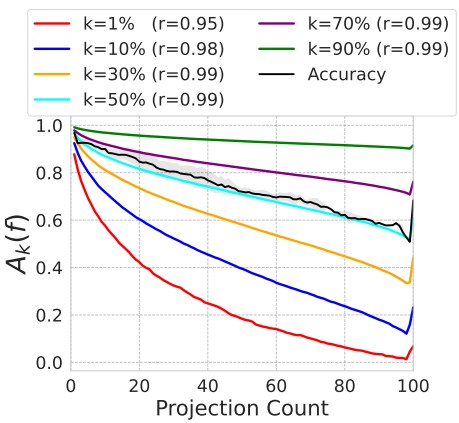

Figure 2: Correlation of alignment score, $A_k(f)$, with accuracy for a synthetic dataset. For different $k$ values, $A_k(f)$ achieve high Pearson correlation ~0.99.

and $A_k$ scores (shown in Figure 2). We find the $A_k$ is highly correlated with the prediction accuracy achieving Pearson correlation scores $\sim 0.99$ (the average correlation over multiple runs). We also compare $A_k$ with a few different alignment measures (including MI estimates) and find that our method outperforms others (more details in Appendix B).

**Lemma 2** (Alignment for random representations). *Expected alignment score achieved by a concept erasure framework $f$ that generates random representations is* $\mathbb{E}[A_k(f)] = k/n$.

The proof is provided in Appendix A.2. This result shows the importance of choosing $k$. If $k$ is too small, the $A_k(f)$ scores may be low for many concept erasure functions. Conversely, if $k \approx n$, then the $A_k(f)$ scores will almost always be close to 1. In our experiments (Figure 2), we find that $A_k$'s correlation is maximized when $k = 0.5n$.

## 5    Evaluation

In this section, we provide the specifics of the experimental setup and evaluation results for concept erasure using KRaM across various datasets. The implementation of KRaM is publicly available at https://github.com/brcsomnath/KRaM.

**Setup**. In all settings, we follow the same routine for concept erasure: (a) we obtain representations either directly from the dataset or an encoder (e.g., BERT, GPT-3), which are kept frozen; (b) we perform concept erasure in a post-hoc manner to obtain representations $f(x) \in \mathcal{Z}$, where $f$ is a non-linear neural network; and (c) we use $f(x)$ on downstream tasks and report the metrics.

**Datasets**. We assess the effectiveness of KRaM in erasing 3 types of concept variables: (a) *categorical concepts* – we apply KRaM to erase binary gender variables from GloVe embeddings and race from BERT embeddings for tweets in the DIAL dataset [7]; (b) *continuous concepts* – we evaluate KRaM on a synthetic dataset, generated using a continuous latent variable, and UCI Crimes [36]. For these datasets, we treat one of the latent continuous variables and African American (AAE) population ratio as the concepts to be erased, respectively; (c) *vector-valued concepts* – we evaluate on Jigsaw toxicity detection dataset [1], where we consider religion and gender (which are vector-valued variables) as the concepts to be erased from GPT-3.5 [15] embeddings. We present additional details in Appendix C.1.

**Baselines**. We compare KRaM with the following baselines: (a) INLP [49] (linear) iteratively projects representations onto the nullspace of optimal separating linear subspaces; (b) RLACE [50] (linear) is an extension of INLP that performs concept erasure by solving minimax game; (c) KCE [51] (non-linear) presents a kernelized version of the minimax game introduced in RLACE; (d) FaRM [18] (non-linear) employs rate-distortion maximization for erasing categorical concepts ; (e) KRaM$_{\text{linear}}$ uses KRaM's with a linear erasure function $f$. To the best of our knowledge, there are no existing methods for continuous or vector-valued concept erasure. For continuous concepts, we normalize the labels and quantize them into $n_b$ bins (a hyperparameter). We denote a concept erasure method as Method$_Q$ whenever quantization is used. For vector-valued concepts, we extend nullspace projection-based

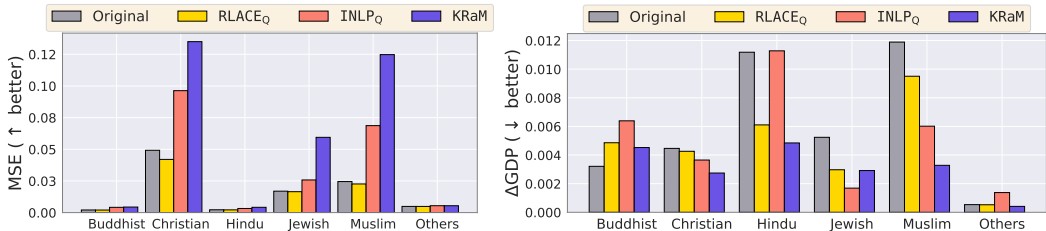

Figure 3: Vector-valued concept (religion) erasure performance using KRaM on Jigsaw toxicity dataset. KRaM achieves better performance ($\uparrow$ MSE & $\downarrow$ $\Delta$GDP) than baseline approaches in most settings.

erasure techniques by quantizing each dimension and projecting onto a series of nullspaces. It is unclear how to utilize non-projection-based methods for vector-valued concept erasure.

**Metrics**. Following previous work [18, 50], we assess concept erasure quality using the following:

*Probing representations*. We use a scikit-learn MLP classifier (non-linear) [45] to probe $f(x)$, report classification accuracy for categorical attributes, MSE for continuous and vector-valued attributes (in a dimension-wise manner). A better concept erasure method would have low accuracy and a high MSE for the deleted concept.

*Downstream metrics*. For datasets with a downstream task, we evaluate if the deleted concept still affects the task by measuring statistical parity. We report DP for categorical concepts, $\Delta$GDP [33] (see Appendix C.2) for continuous concepts, and $\Delta$GDP values for each dimension of vector-valued concepts. Lower statistical parity scores are expected after concept erasure. However, we note that concept erasure does not necessarily guarantee fairness [16]. Applying these methods for fairness would require a more application-specific analysis of the risks and biases.

*Alignment*. We report the alignment scores ($A_k$ for $k = 0.5n$, therefore $A_k \in [0.5, 1]$) wherever a downstream task is absent. Higher $A_k$ scores are expected. If a downstream task is available, we probe $f(x)$ for that task, where we expect high accuracy or low MSE scores.

## 5.1 Main Results

In this section, we report the performance of KRaM in erasing different types of concepts: categorical, continuous, and vector-valued variables. Note that the primary objective of concept erasure is to robustly erase the concept variable (by achieving lower probing and fairness metrics) even if that reduces the utility of the representations to some extent. This is different from adversarial learning where we try to achieve a balance between fairness and utility. In our experiments, we show KRaM is able to robustly erase concepts while retaining a significant amount of original information.

**Vector-valued Concept Erasure.** Erasure of vector-valued concepts is useful when the attribute to be deleted is not available in the form of a categorical or normalized continuous variable. We evaluate KRaM on erasing information about religion, a vector-valued concept, in the Jigsaw toxicity dataset [1], where the downstream task involves detecting whether an online comment is toxic. Each text is annotated with a vector-valued concept: religion with scores over the categories ({'buddhist', 'christian', 'hindu', 'jewish', 'muslim', 'others'}). We obtain text representations from GPT-3.5 API and use a RBF kernel with a cosine distance function. In Figure 3, we report the MSE and $\Delta$GDP scores for KRaM along with other baselines. We observe that KRaM performs the best achieving low $\Delta$GDP with high MSE scores across most concept labels. We also observe that the change in toxicity classification accuracy ($93.2\% \rightarrow 92.1\%$) is minimal during concept erasure. This showcases the efficacy of KRaM in erasing vector-valued attributes as it achieves up to 76% MSE gains over the best baselines. We also report the results for vector-valued gender erasure in Appendix D.

**Continuous Concept Erasure**. We evaluate the efficacy of KRaM in erasing continuous concepts on a synthetic dataset and UCI Crimes. We compare with baseline approaches that use quantized concept labels. In Table 1 (left), we report the results on the synthetic dataset and observe that KRaM performs the best in preventing leakage of $a$ (high MSE scores) while achieving considerable alignment score, $A_k$. While both FaRM and KRaM utilize non-linear erasure functions, a necessity for robust concept erasure, they tend to achieve relatively lower $A_k$ scores. Despite this, it is important to note that significant information can still be preserved through non-linear warping, which we show

| | Synthetic | | | UCI Crimes | | | |
|---|---|---|---|---|---|---|---|
| Method | MSE $(a)$ ↑ | $A_k$ ↑ | Rank ↑ | MSE $(y)$ ↓ | MSE $(a)$ ↑ | ΔGDP ↓ | $A_k$ ↑ |
| Original | 0.006 | 1.0 | 100 | 0.046 | 0.030 | 0.058 | 1.0 |
| Random | 0.174 | 0.50 | 100 | 0.211 | 0.251 | 0.006 | 0.50 |
| INLP$_Q$ [49] | 0.084 🏆 | 0.85 🏆 | 100 | 0.055 🏆 | 0.056 | 0.0 🏆 | 0.90 🏆 |
| RLACE$_Q$ [50] | 0.021 | 0.87 🏆 | 100 | 0.038 🏆 | 0.022 | 0.051 | 0.81 |
| FaRM$_Q$ [18] | 0.068 | 0.74 | 100 | 0.050 🏆 | 0.064 🏆 | 0.013 🏆 | 0.62 🏆 |
| KRaM | 0.109 🏆 | 0.67 | 100 | 0.069 | 0.104 🏆 | 0.001 🏆 | 0.59 |
| KRaM$_{linear}$ | 0.083 🏆 | 0.75 🏆 | 100 | 0.067 | 0.082 🏆 | 0.022 | 0.69 🏆 |

Table 1: Continuous concept erasure: We evaluate on the synthetic and UCI Crimes. Post concept erasure using KRaM, we observe a significant increase in MSE $(a)$ combined with a drop in ΔGDP.

| | DIAL | | | Glove | | |
|---|---|---|---|---|---|---|
| Method | Acc. $(y)$ ↑ | Acc. $(a)$ ↓ | DP ↓ | Acc. $(a)$ ↓ | $A_k$ ↑ | Rank ↑ |
| Original | 75.5 | 87.7 | 0.26 | 100.0 | 1.0 | 300 |
| Random | 50.8 | 50.5 | 0.01 | 50.2 | 0.50 | 300 |
| INLP [49] | 75.1 🏆 | 69.5 | 0.16 | 86.3 | 0.85 🏆 | 210 |
| RLACE [50] | 75.5 🏆 | 82.1 | 0.18 | 95.5 | 0.93 🏆 | 300 🏆 |
| KCE [51] | 75.0 | 80.1 | 0.12 🏆 | 63.5 🏆 | 0.62 | 100 |
| FaRM [18] | 74.8 | 54.2 🏆 | 0.09 🏆 | 53.9 🏆 | 0.65 | 247 🏆 |
| KRaM | 72.4 | 54.0 🏆 | 0.08 🏆 | 52.6 🏆 | 0.65 | 246 🏆 |
| KRaM$_{linear}$ | 75.4 🏆 | 67.5 🏆 | 0.18 | 67.0 | 0.73 🏆 | 130 |

Table 2: Categorical concept erasure: We assess binary gender and race erasure from GloVe and BERT representations (from DIAL) respectively. We denote the top 3 results for any metric using 🏆, 🏆, and 🏆 respectively. Desired trends for all metrics are shown using ↑ or ↓.

in the categorical experiments. Note that linear erasure functions are able to retain nearest neighbour structures better, thereby achieving higher $A_k$ scores. In Figure 5, we visualize the UMAP projection of synthetic data, where the representations' position is indicative of the latent continuous concept attribute prior to concept erasure (left). Post concept erasure (right), we observe no such discernible correlation. For UCI Crimes, we perform erasure for the African-American (AAE) population ratio, and use the generated representations to predict the normalized number of crimes per capita $(y)$. Table 1 (right) shows that KRaM generates representations with minimal information about AAE ratio (high MSE $(a)$) and low ΔGDP scores (ΔGDP $\sim 0$). These experiments showcase KRaM's efficacy in erasing continuous attributes and minimizing their impact on downstream tasks (low ΔGDP scores).

**Categorical Concept Erasure**. In Table 2, we evaluate categorical concept erasure on DIAL tweet classification (race) and GloVe (gender) datasets. For DIAL dataset, we obtain BERT [34] representations of tweets, perform concept erasure for race (binary) attribute, and use the generated representations $f(x)$ for sentiment classification. We report the accuracy of predicting sentiment $(y)$, race $(a)$, and demographic parity (DP) of the predictions in Table 2 (left). We observe that KRaM performs the best in erasing race attribute (evident from high Acc. $(a)$) and demographic parity while attaining comparable accuracy on sentiment classification (Acc. $(y)$). For GloVe embeddings (Table 2 (right)), we observe that KRaM achieves the state-of-the-art result in suppressing the gender leakage (probing accuracy for predicting binary gender attribute). We observe that linear techniques INLP and RLACE, obtain high alignment scores, $A_k$, but their representations still contain significant gender information (indicated by high Acc. $(a)$). This shows a trade-off between information alignment and concept erasure, where robustly erasing a concept may also result in the removal of other information. Categorical concept erasure has been extensively studied, and the fact that a general erasure framework, KRaM, can perform on par with state-of-the-art methods underscores its efficacy.

## 5.2 Analysis

In this section, we perform several analysis experiments to understand the functioning of KRaM.

**Comparison with MI estimation techniques**. In this experiment, we compare our concept erasure framework with a few state-of-the-art mutual information (MI) estimation approaches. Essentially,

| Dataset | Acc. $(a) \downarrow$ | $A_k \uparrow$ | WS-353 $\uparrow$ |
|---|---|---|---|
| GloVe | 100.0 | 1.0 | 0.70 |
| InfoNCE [43] | 64.0 | 0.55 | 0.19 |
| MINE [9] | 76.8 | 0.54 | 0.03 |
| CLUB [17] | 98.1 | 0.55 | 0.06 |
| KNIFE [47] | **50.2** | 0.53 | 0.10 |
| KRaM | 52.6 | **0.65** | **0.48** |

Figure 4: Comparison of KRaM with state-of-the-art mutual information (MI) estimation methods. KRaM achieves a balance between good concept erasure nearly with high $A_k$.

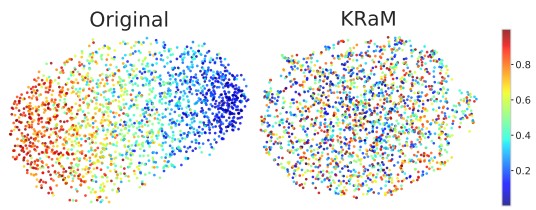

Figure 5: Visualization of UMAP projection of representations obtained from the synthetic dataset before and after concept erasure. Concept erasure makes the positions of representations uncorrelated with their continuous concept labels (denoted by their color).

the task of concept erasure can be formalized as maximizing the objective: $I(\mathcal{Z}, \mathcal{X}) - I(\mathcal{Z}, \mathcal{A})$, where $I(\cdot, \cdot)$ denotes the mutual information between two sets. We optimize this objective function with the following MI estimates: InfoNCE [43], MINE [9], CLUB [17], and KNIFE [47]. In Table 4, we report the gender accuracy (Acc. $(a)$), $A_k$, and performance on WordSim-353 benchmark [2], which is also reflective of the alignment, on GloVe dataset. We observe that MI techniques lose significant information from the original representations achieving near-random $A_k$ and WS-353 scores. We believe this happens because MI approaches optimize their estimation parameters along with the erasure function $f$, which is difficult.

We observe a unique scenario for the CLUB method, where the Acc. $(a)$ is high and $A_k$ is low. This implies that the clusters (related to different genders) may be retained but the nearest neighbour structure within the clusters is perturbed significantly. Probing for the downstream task (Acc. $(a)$) may give you the impression that information from the original space is retained, while $A_k$ provides a more fine-grained view contradicting such an incorrect conclusion. In general, we find that $A_k$ values are relatively well correlated with WS-353 scores, which is a good measure of the information retained in the representation space after erasure. However, computing WS-353 requires additional annotation that may not be feasible for representation sets other than word embeddings.

We also compare KRaM with several other mutual information estimation based baselines that use a unique objective function for controlling information in representations. Specifically, we compare with MIFR [55], CCL [56], and ICVAE [42] and report the results on DIAL dataset in Table 3. Note that all of these methods work for categorical concepts only, and ICVAE requires access to concept labels for test instances as well, a limitation compared to KRaM and the other methods. In Table 3, we observe MIFR is unable to erase concepts robustly (based on Acc. $(a)$ and DP scores). ICVAE and CCL are able to delete concepts but at the significant cost of deleting a lot of original

| Method | Acc $(y) \uparrow$ | Acc $(a) \downarrow$ | DP $\downarrow$ |
|---|---|---|---|
| Original | 75.5 | 87.7 | 0.26 |
| MIFR [55] | 75.4 | 68.7 | 0.21 |
| CCL [56] | 50.7 | 52.6 | 0.01 |
| ICVAE [42] | 66.5 | 53.3 | 0.10 |
| KRaM | 72.4 | 54.0 | 0.08 |

Table 3: Comparison of KRaM with mutual information based baseline approaches on DIAL dataset. We observe that KRaM achieves a fine balance between concept erasure and retaining task performance.

information. Loss of information from the original space (low Acc. $(y)$) using ICVAE and CCL is quite similar to other MI methods reported in Figure 4. Compared to these methods, KRaM is able to get similar concept erasure performance while achieving much higher Acc. $(y)$ scores. We report additional results using these baselines on the GloVe dataset in Appendix D.

**Image-based datasets**. We perform experiments on image-based datasets to evaluate the efficacy of KRaM on different domains. We consider two different setups using CelebA [37] and Colored MNIST [6] datasets. In CelebA, we consider the binary variable *attractiveness* as the target attribute and whether the face has makeup applied (binary variable) as the protected attribute. With concept erasure using KRaM, the accuracy of predicting the makeup attribute dropped from 85.7% to 69.6%. The demographic parity of the predictions (for attractiveness) also improved from 0.94 to 0.54. This shows the effectiveness of KRaM in removing the makeup concept.

For Colored MNIST, we follow the setup of [6] to create a biased version of the MNIST dataset [22]. The background color of the digits is made to be well correlated with the digit in the training set. However, no such correlation exists in the test set. We use a 0.8 correlation between the background

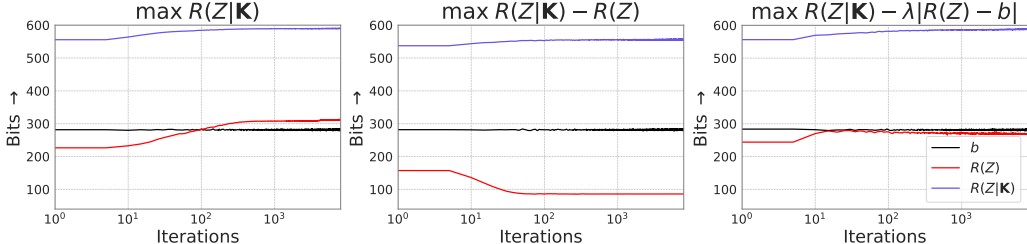

Figure 6: Evolution of $R(\mathcal{Z})$ and $R(\mathcal{Z}|\mathbf{K})$ under different setups. We monitor the impact on the volume of the overall feature space in different settings of the objective function.

color and the digit label. We treat background color as the concept to be erased. Naively training a classifier on the images will perform poorly on the test set as the classifier can easily overfit the background color. The digit information could be predicted with an accuracy of 79.9%. Post-concept erasure we observe that the classifier's performance improves from 79.9% to 87.1%. This shows the effectiveness of KRaM to remove background information and help the classifier generalize better.

**Evolution of loss functions**. In this experiment, we investigate the evolution of loss terms $R(\mathcal{Z})$ and $R(\mathcal{Z}|\mathbf{K})$ under various settings of the objective function (described in Figure 1) on the synthetic dataset. In Figure 6 (a), we observe that indeed maximizing the kernelized rate-distortion function also leads to an increase in $R(\mathcal{Z})$, which can result in the concept variable not being completely erased. In Figure 6 (b), we examine the scenario where $R(\mathcal{Z}|\mathbf{K}) - R(\mathcal{Z})$ is maximized, where we observe indicate a substantial drop in $R(\mathcal{Z})$. This reduction often implies a low intrinsic dimension and a substantial information loss. In Figure 6 (c), we present the evolution of loss terms for KRaM's objective. We notice that $R(\mathcal{Z})$ aligns closely with the initial number of bits, denoted as $b$. Meanwhile, the $R(\mathcal{Z}|\mathbf{K})$ is maximized to its full extent (similar to Figure 6 (a)). Through these experiments, we provide empirical verification for the insights discussed in relation to Figure 1.

We conduct additional ablation experiments that involve varying hyperparameters and kernel functions. The results of these experiments are reported in Appendix D.

## 6 Conclusion

In this paper, we proposed KRaM, a novel framework to robustly perform concept erasure from a representation set. KRaM uses a kernelized formulation of the rate-distortion function, where the kernel is created using concept labels. This approach ensures that instances with similar concept labels become dissimilar in the representation space, which ultimately results in the erasure of the concept variable. KRaM is a versatile method capable of erasing a wide range of concepts, including categorical, continuous, and vector-valued variables. We theoretically analyze several properties of the proposed KRaM objective. Empirical evaluation shows the efficacy of KRaM on a wide range of setups ranging from gender erasure from GloVe embeddings to vector-valued concept erasure from GPT-3.5 embeddings. We also propose a heuristic-based measure to capture the information alignment of the erasure function $f$ by analyzing the $k$-nearest neighbours of the representations. While KRaM effectively erases concepts, it does result in the loss of some information from the original space, as evidenced by the alignment scores. Determining the minimum amount of information that must be distorted to fully erase a concept remains an open question. Future research could concentrate on gaining a deeper understanding of this issue and developing techniques that can erase concepts from representations while having minimal impact on alignment with original representations.

#### Acknowledgement

The authors are thankful to Alex Beutel (OpenAI previously Google Research) and Kelsey Allen (Google DeepMind) for helpful feedback on an earlier version of this paper. The work of Somnath Basu Roy Chowdhury and Snigdha Chaturvedi was supported in part by Amazon Research Awards.

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

# A  Theoretical Proofs

## A.1  Proof of Lemma 1

We prove the general bounds for $R(\mathcal{Z}|\mathbf{K})$ by independently proving the lower and upper bound using the following intermediate results.

**Lemma 3** (Lower bound for $R(\mathcal{Z}|\mathbf{K})$)**.** *For any* $\mathcal{Z} \in \mathbb{R}^{n \times d}$, *a kernel matrix* $\mathbf{K} \in \mathbb{R}^{n \times n}$ *using a kernel function satisfying* $k(x,x) = 1$ *and* $\epsilon > 0$, *it holds that:*

$$R(\mathcal{Z}|\mathbf{K}) \geq R(\mathcal{Z}), \tag{7}$$

*where the equality is satisfied only when* $\mathbf{K} = \mathbf{1}\mathbf{1}^T$.

*Proof.* We start off by writing down the expanded form of $R(\mathcal{Z}|\mathbf{K})$ as:

$$R(\mathcal{Z}|\mathbf{K}) = \frac{1}{2}\log_2 \det\left(I + \frac{d}{n\epsilon^2}\mathcal{Z}\mathcal{Z}^T \odot \mathbf{K}\right). \tag{8}$$

In the above equation 8, we first note that both $\mathcal{Z}\mathcal{Z}^T$ and $\mathbf{K}$ matrices are positive-semi definite symmetric matrices. Using Schur product theorem [54], we can show that their hadamard product $\mathcal{Z}\mathcal{Z}^T \odot \mathbf{K}$ is also positive semi-definite (for $d > 1$). Next, we utilize the following property for Hadamard products:

*Theorem 7.25* [53]. Given two positive semi-definite square matrices $A$ and $B$ of dimension $m$. Then, the following property holds: $\det(A \odot B) \geq \det(A) \prod\limits_{i=1}^{m} b_{ii}$

Applying this property to $\mathcal{Z}\mathcal{Z}^T \odot \mathbf{K}$, we get the following result:

$$\det(\mathcal{Z}\mathcal{Z}^T \odot \mathbf{K}) \geq \det(\mathcal{Z}\mathcal{Z}^T), \tag{9}$$

where $\mathbf{K}_{ii} = 1, \forall i$. Now, since $\mathcal{Z}\mathcal{Z}^T \odot \mathbf{K}$ and $\mathcal{Z}\mathcal{Z}^T$ are positive semi-definite, their corresponding eigenvalues are non-negative, $\lambda_i(\mathcal{Z}\mathcal{Z}^T \odot \mathbf{K}) \geq 0$ and $\lambda_i(\mathcal{Z}\mathcal{Z}^T) \geq 0$. Since the eigenvalues are non-negative, we can extend Equation 4 as follows:

$$\prod_i \lambda_i(\mathcal{Z}\mathcal{Z}^T \odot \mathbf{K}) \geq \prod_i \lambda_i(\mathcal{Z}\mathcal{Z}^T)$$

$$\prod_i \left(1 + \frac{d}{n\epsilon^2}\lambda_i(\mathcal{Z}\mathcal{Z}^T \odot \mathbf{K})\right) \geq \prod_i \left(1 + \frac{d}{n\epsilon^2}\lambda_i(\mathcal{Z}\mathcal{Z}^T)\right) \tag{10}$$

$$\det\left(I + \frac{d}{n\epsilon^2}\mathcal{Z}\mathcal{Z}^T \odot \mathbf{K}\right) \geq \det\left(I + \frac{d}{n\epsilon^2}\mathcal{Z}\mathcal{Z}^T\right)$$

$$R(\mathcal{Z}|\mathbf{K}) \geq R(\mathcal{Z}),$$

where the second inequality holds because the affine transform of positive variables preserves inequalities. The equality is satisfied when $\mathbf{K} = \mathbf{1}\mathbf{1}^T$. □

**Lemma 4** (Upper bound for $R(\mathcal{Z}|\mathbf{K})$)**.** *For any* $\mathcal{Z} \in \mathbb{R}^{n \times d}$, *any kernel matrix* $\mathbf{K} \in \mathbb{R}^{n \times n}$ *using a kernel function satisfying* $k(x,x) = 1$ *and* $\epsilon > 0$, *it holds that:*

$$R(\mathcal{Z}|\mathbf{K}) \leq \frac{n}{2}\log_2\left(1 + d/n\epsilon^2\right). \tag{11}$$

*Proof.* We start by noting that the Hadamard product of two positive semi-definite matrices $\mathcal{Z}\mathcal{Z}^T \odot \mathbf{K} \in \mathbb{R}^{n \times n}$ is positive semi-definite (using the Schur product theorem). We also assume that the representations $z_i \in \mathcal{Z}$ are unit normalized, thereby the diagonal entries of $(\mathcal{Z}\mathcal{Z}^T)_{ii} = 1$. The diagonal entries $\mathbf{K}_{ii} = 1$, which implies $(\mathcal{Z}\mathcal{Z}^T \odot \mathbf{K})_{ii} = 1$. Given these facts, we can write the following properties of about the eigenvalues of $\mathcal{Z}\mathcal{Z}^T \odot \mathbf{K}$:

$$\lambda_i(\mathcal{Z}\mathcal{Z}^T \odot \mathbf{K}) \geq 0, \ \sum_i \lambda_i(\mathcal{Z}\mathcal{Z}^T \odot \mathbf{K}) = n, \tag{12}$$

where the second property follows from the fact that $\mathrm{tr}(\mathcal{Z}\mathcal{Z}^T \odot \mathbf{K}) = n$. We are interested in finding the maximum value of $R(\mathcal{Z}|\mathbf{K})$ that can be written as:

$$R(\mathcal{Z}|\mathbf{K}) = \frac{1}{2}\log_2 \prod_{i=1}^{n}\left(1 + \frac{d}{n\epsilon^2}\lambda_i(\mathcal{Z}\mathcal{Z}^T \odot \mathbf{K})\right). \tag{13}$$

To maximize $R(\mathcal{Z}|\mathbf{K})$, we need to maximize the product within the logarithm. Each term within the product $1 + \frac{d}{n\epsilon^2}\lambda_i(\mathcal{Z}\mathcal{Z}^T \odot \mathbf{K}) \geq 0$ (eigenvalues of a PSD matrix). Using the AM-GM inequality, the product is maximized when all the individual terms are equal,

$$\lambda_i(\mathcal{Z}\mathcal{Z}^T \odot \mathbf{K}) = n/n = 1. \tag{14}$$

Substituting this result in Equation 13, we obtain the following upper bound:

$$R(\mathcal{Z}|\mathbf{K}) \leq \frac{n}{2}\log_2(1 + d/n\epsilon^2), \tag{15}$$

where the equality is achieved when $\mathcal{Z}\mathcal{Z}^T = I$ when all the representations are orthogonal. Note that this is only possible when $d \geq n$.

$\square$

*Proof of Lemma 1.* By combining the results of Lemma 3 & 4, we get the following:

$$R(\mathcal{Z}) \leq R(\mathcal{Z}|\mathbf{K}) \leq \frac{n}{2}\log_2(1 + d/n\epsilon^2). \tag{16}$$

This completes the proof. $\square$

## A.2 Proof of Lemma 2

**Lemma 2** (Alignment for random representations). *Expected $A_k(f)$ score achieved by a concept erasure framework $f$ that generates random representations is $\mathbb{E}[A_k(f)] = k/n$.*

*Proof.* To prove this, we first assume two randomly generated $k$-nearest neighbour graphs (since the original representation is uncorrelated with the randomly generated one we can consider it as random). As it is a $k$NN graph, for each node has an expected degree $\mathbb{E}[d] \approx k$, where $d$ is the degree of the node. Now, let's consider the probability of a node $x_i$ being part of node $x_j$:

$$p(x_i \in \mathrm{knn}(x_j)) = \frac{d_i}{n}$$
$$\mathbb{E}[p(x_i \in \mathrm{knn}(x_j))] = \frac{\mathbb{E}[d_i]}{n} = \frac{k}{n}, \tag{17}$$

where $d_i$ is the degree of node $i$ and $n$ is the total number of representations. Since computing the exact probability requires knowledge of the degree of the node, we compute the expectation of the same. Next, we compute the probability that node $i$ is present in both $k$NN sets (before and after debiasing) of node $j$:

$$\begin{aligned}
\mathbb{E}[\mathrm{knn}(x) \cap \mathrm{knn}(z)] &= \mathbb{E}\left[\sum_j p(x_i \in \mathrm{knn}(x_j) \wedge z_i \in \mathrm{knn}(z_j))\right] \\
&= \sum_j \mathbb{E}\left[p(x_i \in \mathrm{knn}(x_j))p(z_i \in \mathrm{knn}(z_j))\right] \\
&= \sum_j \mathbb{E}\left[p(x_i \in \mathrm{knn}(x_j))\right]\mathbb{E}\left[p(z_i \in \mathrm{knn}(z_j))\right] \\
&= \sum_{j=1}^{n} k^2/n^2 = k^2/n,
\end{aligned} \tag{18}$$

where the first step utilizes linearity of expectation, and the second step follows from the fact that the degree of distribution of $\mathcal{X}$ and $\mathcal{Z}$ are independent. Replacing the result from Eqn 18 in Eqn 6, we get $\mathbb{E}[A_k(f)] = k/n$. $\square$

### A.3 Additional Theoretical Results

**Lemma 3** (Upper Bound of $R(\mathcal{Z}|\mathbf{K})$ for categorical concepts). *For categorical concept variables with the kernel values $\mathbf{K}_{ij} \in \{0,1\}$, $R(\mathcal{Z}|\mathbf{K})$ is bounded by the sum of rate-distortion functions of representation set from individual classes $\mathcal{Z}_j$*

$$R(\mathcal{Z}|\mathbf{K}) = \sum_{j=1}^{m} \frac{1}{2} \log_2 \det \left( I + \frac{d}{n\epsilon^2} \mathcal{Z}_j \mathcal{Z}_j^T \right) \leq \sum_{j=1}^{m} R(\mathcal{Z}_j), \tag{19}$$

*where the equality holds only when $\mathcal{Z}_j \mathcal{Z}_j^T = 0, \forall j$ and $m$ is the number of classes.*

*Proof.* For categorical variables, the kernel function takes the following form:

$$k(i,j) = \begin{cases} 1, & \text{if } a_i = a_j \\ 0, & \text{if } a_i \neq a_j \end{cases}. \tag{20}$$

If the kernel function $k(\cdot, \cdot)$ is of the above form. Using the corresponding kernel matrix $\mathbf{K}$ we get,

$$\mathbf{M} = I + \frac{d}{n\epsilon^2} \mathcal{Z}\mathcal{Z}^T \odot \mathbf{K} = I + \frac{d}{n\epsilon^2} \begin{bmatrix} \mathcal{Z}_1 \mathcal{Z}_1^T & 0 & \dots & 0 \\ 0 & \mathcal{Z}_2 \mathcal{Z}_2^T & \dots & 0 \\ \vdots & \vdots & \ddots & \vdots \\ 0 & 0 & \dots & \mathcal{Z}_k \mathcal{Z}_k^T \end{bmatrix}, \tag{21}$$

where $\mathbf{M}$ becomes a block diagonal matrix and $Z_i$'s are representations belonging to class $i$. Using the determinant property of block diagonal matrices, we have:

$$\log_2 \det(\mathbf{M}) = \sum_{j=1}^{m} \log_2 \det \left( I + \frac{d}{n\epsilon^2} \mathcal{Z}_j \mathcal{Z}_j^T \right)$$
$$R(\mathcal{Z}|\mathbf{K}) = \sum_{j=1}^{m} \frac{1}{2} \log_2 \det \left( I + \frac{d}{n\epsilon^2} \mathcal{Z}_j \mathcal{Z}_j^T \right). \tag{22}$$

The individual terms in the above summation are closely related to the rate-distortion function of representation belonging to each class, $j$, as shown below:

$$R(\mathcal{Z}_j) = \frac{1}{2} \log_2 \det \left( I + \frac{d}{n_j \epsilon^2} \mathcal{Z}_j \mathcal{Z}_j^T \right), \tag{23}$$

where $n_j$ is the number of representations in class $j$. Note, $n_j < n$, where $n$ is the total number of representations. Using the property that multiplying a matrix with a scalar is equivalent to multiplying its eigenvalues with the same scale, and that $\mathcal{Z}_j \mathcal{Z}_j^T$ is a PSD matrix. We can show:

$$R(\mathcal{Z}|\mathbf{K}) \leq \sum_{j=1}^{m} \frac{1}{2} \log_2 \det \left( I + \frac{d}{n\epsilon^2} \mathcal{Z}_j \mathcal{Z}_j^T \right)$$
$$R(\mathcal{Z}|\mathbf{K}) \leq \sum_{j=1}^{m} R(\mathcal{Z}_j). \tag{24}$$

This completes the proof. $\qquad\square$

**Discussion**. Notice that this is closely related to the MCR$^2$ objective, which tries to learn discriminative subspaces for individual classes. For concept erasure, we aim for the opposite effect by making instances from the same class dissimilar by maximizing their rate-distortion function.

---

**Algorithm 1** Correlation Computation Routine

---

1: **Input**: Input representation set $\mathcal{X} \in \mathbb{R}^{n \times d}$
2: $\mathcal{Y} = \text{sgn}(\mathcal{X}W_1W_2)$       ▷ generate labels using random weights $W_2 \in \mathbb{R}^{d \times m}, W_1 \in \mathbb{R}^{m \times 1}$
3: $\mathbf{U}, \Sigma, \mathbf{V} = \text{svd}(\mathcal{X})$
4: $\mathcal{Z}_0 = \mathcal{X}$       ▷ Initializing the representations
5: $A = \{\}, \text{scores} = \{\}$       ▷ accuracy and alignment sets
6: **for** $i \in \{1, \dots, d\}$ **do**
7:      $\mathbf{u} = \frac{\mathbf{V}^T(i)}{\|\mathbf{V}^T(i)\|}$       ▷ access the $i$-th column of $\mathbf{V}$
8:      $\mathbf{P}_i = \mathbf{I}_d - \mathbf{u}\mathbf{u}^T$       ▷ null space projection matrix
9:      $\mathcal{Z}_i = \mathcal{Z}_{i-1}\mathbf{P}_i$
10:      $A = A \cup \text{acc}(\mathcal{Z}_i, \mathcal{Y})$       ▷ compute accuracy
11:      $\text{scores} = \text{scores} \cup A_k(\prod \mathbf{P}_i)$       ▷ compute alignment scores
12: **end for**
13: $r = \text{Pearson}(A, \text{scores})$       ▷ compute the Pearson correlation
14: **return** $r$

---

## B    Alignment Scoring

In this section, we present several measures to capture information alignment and compare them with our proposed metric (in Section 4).

**KSG MI estimator** [35]. The Kraskov–Stogbauer–Grassberger (KSG) estimator uses the nearest neighbour information in the joint and marginal space to obtain a mutual information estimate. Specifically, it computes the number of neighbours around a point within a hypercube in the marginal spaces. The length of the hypercube is set based on the max-norm distance to the $k$-th neighbour in the joint space. The KSG MI estimate between two sets $\mathcal{X}$ and $\mathcal{Z}$ can be shown as follows:

$$I_{\text{KSG}}(\mathcal{X}, \mathcal{Z}) = \psi(k) - 1/k - \mathbb{E}[\psi(n_x) + \psi(n_z)] + \psi(N), \tag{25}$$

where $\psi(\cdot)$ is the digamma function, $n_x$ and $n_z$ are the number of points in the hypercube of the respective marginal spaces. In our experiment, we use the KSG MI estimator to evaluate the alignment between representation sets before and after concept erasure.

**Degree distribution**. In a $k$-nearest neighbour graph, some nodes are more connected to others (hub nodes) while others are sparsely connected. Building on our intuition of alignment $A_k$ using the nearest neighbour graphs of representations, we can consider changes in its degree distribution, $D(\mathcal{X})$, during concept erasure to gauge how the underlying structure of the representation set has changed. We quantify the change using either L1-norm, L2-norm, or KL-divergence between the normalized degree distributions $D(\mathcal{X})$ and $D(\mathcal{Z})$.

**Experiments**. We perform experiments in a controlled setup to evaluate the efficacy of the proposed alignment measures.

(a) *Simulated Erasure*. In this experiment, we simulate knowledge erasure from a set of synthetic representations and observe how the alignment scores correlate with the downstream accuracy. Algorithm 1 shows the details for this process. First, we sample a set of representations from a uniform distribution $\mathcal{X} \sim \mathbb{R}^{n \times d}$ from a uniform distribution and construct a label set $\mathcal{Y}$ (using randomly sampled weights $W_1, W_2$). In a way, the label set retains some information about the original representations that we will probe as erasure happens. Then, we gradually remove information from representations $\mathcal{Z}$ by projecting them onto the nullspace $\mathbf{P}$ formed using the eigenvectors $\mathbf{u}$. After each iteration of projection, we compute the accuracy of predicting $\mathcal{Y}$ and alignment score, $A_k$. We report the Pearson correlation between the accuracies and information alignment in Table 4 (left side), along with the hyperparameter $k$ used for each measure. We observe that $A_k$ outperforms other approaches achieving better correlation, which showcases the efficacy of our approach.

(a) *Correlated Gaussians*. In this experiment, we sample two sets of Gaussians (zero mean) with a fixed covariance $\sigma$. In this setup, the mutual information has a closed-form solution:

$$I(\mathcal{X}, \mathcal{Z}) = -\frac{1}{2}\log(1 - \sigma^2). \tag{26}$$

We use the samples to compute the different alignment measures and investigate if they're correlated with the actual mutual information (Equation 26). Note that there does not exist an explicit mapping

|  | Simulated Erasure | | Correlated Gaussian | |
| Metric | $k/n$ (%) | Pearson $(r)\uparrow$ | $k/n$ (%) | Pearson $(r)\uparrow$ |
| --- | --- | --- | --- | --- |
| KSG | 10 | 0.965 | 0.02 | **0.989** |
| KL-div (degree) | 0.1 | 0.874 | 0.2 | 0.490 |
| L2-norm (degree) | 0.1 | 0.865 | 0.2 | 0.458 |
| L1-norm (degree) | 0.1 | 0.905 | 0.2 | 0.564 |
| Alignment: $A_k$ | 50 | **0.994** | 50 | 0.969 |

Table 4: Comparison of $A_k$ with other alignment measures on synthetic datasets. We observe that $A_k$ achieves the best Pearson correlation scores with downstream accuracy on simulated concept erasure experiments due to the presence of a mapping function $f$. In a separate experiment, the KSG estimator shows the highest correlation with MI. $A_k$ also achieves a high correlation score, while the degree distribution-based measures perform poorly due to the lack of a mapping function.

between these samples. In Table 4 (right side), we report the Pearson correlation scores for different measures. We find that the KSG MI estimator outperforms others, with $A_k$ coming in as a close second. This is because our alignment scores assume a 1-to-1 mapping between the sets, which is absent in this case. The degree-distribution-based scores suffer even more as their measure is even more strongly reliant on the mapping. These results show that the alignment score $A_k$ leverages the bijective mapping to generate scores that are well correlated with the mutual information but can be inaccurate in cases where the mapping function is absent.

## C Implementation details

In this section, we provide various implementation details about our experimental setup. Specifically, we describe the details of the dataset, metrics, and hyperparameters utilized.

### C.1 Dataset

In this section, we describe the details of the datasets that were used in the experimental section.

**GloVe embeddings** [46]. We revisit the problem of deleting gender information (*binary attribute*) from word embeddings [13]. Specifically, we consider the GloVe embeddings of the 150k most frequently occurring words. For training KRaM, we follow the setup of [49, 18] to select the 7500 most male-biased, female-biased, and neutral words determined by the magnitude of the word vector's projection onto the gender direction (the largest principal component of the space of vectors formed using the difference gendered word vector pairs).

**DIAL** [12] is a Twitter-based sentiment classification dataset, where each tweet is associated with sentiment labels and "race" information (binary concept label) of the author. The sentiment concept labels are "happy" or "sad" and the binary race concept labels are "African-American English" (AAE) or "Standard American English" (SAE).

**Synthetic dataset**. We create a dataset where the representations are generated using a continuous latent variable, $a$, which serves as our concept label. During data generation, we first sample the latent variable $a \sim \text{Uni}(0, 1)$, and then sample the high-dimensional representation $x \sim \mathcal{N}(a\mathbf{1}_d, aI_d)$, where $\mathbf{1}_d$ is a vector of ones and $I_d$ is the identity matrix. For this dataset, we set the dimension of the representations to be $d = 100$. In this setup, we observe that the latent concept label, $a$, is being used to scale the underlying isotropic Gaussian distribution. Therefore, post-concept erasure the representation space should appear like an isotropic Gaussian distribution, which is indeed the case as shown in Figure 5.

**UCI Crimes** [36]. This dataset[1] contains information about US communities in 1990 from various surveys. The dataset provides 128 attributes (both categorical and continuous variables) from 1,994 different US communities. we concatenate individual attributes of a community to obtain its representation. The regression task involves predicting the number of violent crimes per capita. We consider the ratio of African-American (AAE) people (*continuous* attribute) in a community as the concept to be erased.

**Jigsaw Toxicity Classification** [1]. This dataset contains online comments and the binary classification task involves detecting whether a comment is toxic or not. In this dataset, we consider two different concepts: *religion* and *race*. We consider a vector-valued protected attribute for this dataset. For the religion concept, we consider an unnormalized vector over the following categories: {'buddhist', 'christian', 'hindu', 'jewish', 'muslim', 'other_religion'}. Similarly, for the gender we consider the following categories: {'bisexual', 'female', 'heterosexual', 'homosexual, gay, or lesbian', 'male', 'other gender', 'other sexual orientation', 'transgender'}. During concept erasure of either concept, we only consider instances where at least one of the concept categories has a non-zero value and reserved 20% of the instances as the test set. This resulted in a dataset with a train/test split of (72k, 18k) for the religion concept and (106k, 26k) for the gender concept. We retrieve text representations for the comments from GPT-3.5 [15] and perform concept erasure on them.

### C.2 Metrics

In this section, we present the details of the fairness metrics used in our experiments.

**Demographic Parity (DP).** Demographic Parity measures the difference in the probability of a prediction w.r.t to the protected attribute $\mathcal{A}$. Formally, it is defined as:

$$\text{DP} = \sum_{y \in \mathcal{Y}} |p(\hat{y} = y | \mathcal{A} = a) - p(\hat{y} = y | \mathcal{A} = \bar{a})|, \tag{27}$$

where $a, \bar{a}$ are possible values of the binary concept and $\mathcal{Y}$ is the set of possible target attribute labels.

---

[1]https://archive.ics.uci.edu/ml/datasets/Communities+and+Crime

**Generalized Demographic Parity ($\Delta$GDP)**. Most of the literature on fairness metrics has focused on categorical variables. We use Generalized Demographic Parity (GDP) [33], which measures the discrepancy in outcome with respect to a continuous variable. GDP measure extends Demographic Parity for continuous protected attributes. It is defined as follows:

$$\Delta\text{GDP} = \int_0^1 |m(a) - m_{\text{avg}}| P(\mathcal{A} = a) da, \tag{28}$$

where $m(a) = \mathbb{E}[\hat{y}|\mathcal{A} = a]$ is expected prediction of the model when protected attribute $\mathcal{A} = a$, $m_{\text{avg}} = \mathbb{E}[\hat{y}]$ is overall expected prediction, and $P(\mathcal{A} = a)$ is the probability that the protected attribute takes value $a$. The probability density $P(\cdot)$ can be measured using a histogram or kernel method. We used a kernel function to evaluate the probability density.

## C.3 Hyperparameters

In our experiments, we primarily deal with two hyperparameters: regularization constant, $\lambda$ (in Equation 4), and $\sigma$, associated with the standard deviation of a Gaussian kernel ($k(x, y) = e^{-\|x-y\|/\sigma^2}$). We set these parameters by performing a grid search on the development set using Weights & Biases [11]. We use a multi-layer neural network with ReLU non-linearity as the erasure function $f$. We further perform ablation experiments to understand the impact of these parameters on concept erasure performance (shown in Figure 8). All networks were trained using a single 22GB NVIDIA Quadro RTX 6000 GPU and experiments were executed in PyTorch [44] framework.

# D Additional Results

In this section, we present additional experiments to analyze KRaM's concept erasure performance.

**Vector-valued Concept Erasure**. In this section, we present the results of vector-valued gender concept removal from GPT-3.5 text embeddings from the Jigsaw Toxicity classification dataset. We report the MSE and $\Delta$GDP results in Figure 7. We observe that KRaM is able to significantly increase the gender MSE while simultaneously reducing the $\Delta$GDP scores. During the debiasing process, we observe that there is minimal impact on the toxicity classification accuracy (91.9% $\rightarrow$ 90.1%).

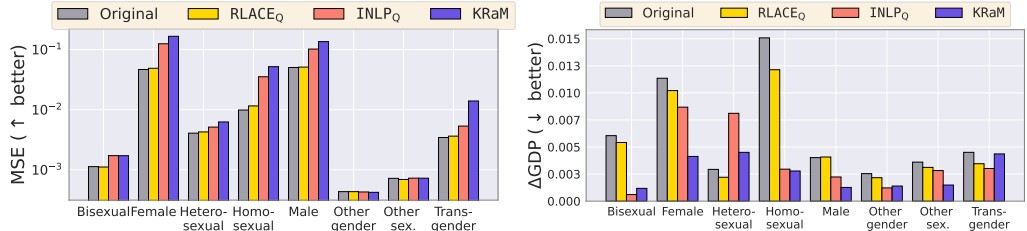

Figure 7: Vector-valued concept erasure performance using KRaM on Jigsaw toxicity classification dataset (gender concept). We observe a significant reduction in $\Delta$GDP scores post erasure of vector-valued gender concept with negligible impact on toxicity classification performance.

**Ablation with different kernels**. We perform ablation experiments with different kernel functions used to define the $\mathbf{K}$ and observe its impact on the concept erasure performance. In Table 5, we report the results for erasing the continuous concept on the synthetic dataset. Apart from the kernel function, we use the same hyperparameters in all setups. We observe that KRaM achieves similar concept erasure performance using different kernel functions. We observe that using the Gaussian kernel function in KRaM yields the best erasure performance and alignment score $A_k$ improves when we use a linear erasure function $f$.

| Method | MSE $(a)\uparrow$ | $A_k\uparrow$ |
|---|---|---|
| Original | 0.006 | 1.0 |
| KRaM (Laplace) | 0.083 | 0.68 |
| KRaM (Cauchy) | 0.092 | 0.63 |
| KRaM$_{\text{linear}}$ (Gaussian) | 0.083 | **0.75** |
| KRaM (Gaussian) | **0.109** | 0.67 |

Table 5: Ablations with kernel functions: we observe that KRaM achieves similar performance using different kernel functions.

**Controlled experiments**. In order to better understand concept erasure performance, we evaluate KRaM on axis-aligned data in a controlled setup. We generate a synthetic two-dimensional dataset involving two Gaussians centered at (0, 2) and (0, -2) respectively, and identity covariance. We consider the $y$-axis as the continuous concept to be deleted. The desired outcome of deleting an axis-aligned concept in two-dimensional data is that points should lie on a 1D line and the KRaM output should be least correlated with the axis concept that was deleted. That is a reduction in dimension such that most of the data variation should be constrained along a single dimension for this case. We looked at the eigenvalues of both the original data and the concept deleted (by KRaM) output. The fraction of eigenvalue masses in the original data (65%, 35%) and in the KRaM concept deleted output the fraction of masses is (99.998%, 0.002%). This shows that KRaM is able to effectively delete the target concept resulting in a drop in intrinsic data dimension.

**Comparison with MI baselines**. We present the results on mutual information (MI) estimation-based baselines on the GloVe dataset. Consistent with the results in Table 3, we observe that other MI estimation baselines are either unable to erase concepts robustly (MIFR) or end up erasing a significant amount of information along with the concept (CCL and ICVAE) in Table 6. In contrast to these approaches, we observe that our rate-distortion based framework, KRaM, is able to achieve a good balance between concept erasure and retaining original information (indicated by $A_k$).

| Method | Acc $(a)\downarrow$ | $A_k\uparrow$ |
|---|---|---|
| Original | 100.0 | 1.0 |
| MIFR [55] | 68.3 | 0.58 |
| CCL [56] | 56.9 | 0.56 |
| ICVAE [42] | 51.5 | 0.50 |
| KRaM | 52.6 | 0.65 |

Table 6: Comparison of KRaM with MI-based baselines on GloVe dataset. We observe that KRaM achieves a good balance between concept erasure and retaining original information.

**Ablation of parameters**. In this experiment, we perform ablations with several parameters in KRaM and observe how that affects the concept erasure performance. First, we ex-

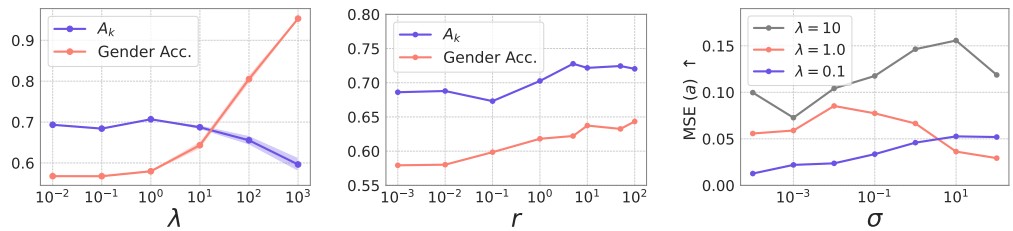

Figure 8: Ablation experiments to study the effect of parameters $\lambda$ (Eqn. 4), $r$ (a scaling factor in $R(\mathcal{Z}) = rb$), and $\sigma$ (parameter in gaussian kernels) on the performance of concept deletion.

periment with gender removal from GloVe embeddings to understand the impact of $\lambda$ (Eqn. 4). In Figure 8 (left), we observe that as $\lambda$ increases, concept erasure worsens ($\uparrow$ gender accuracy). This is expected as the erasure function $f$ is penalized for $|R(\mathcal{Z}) - b|$ term more than maximizing $R(\mathcal{Z}|\mathbf{K})$ (which helps in erasure). The alignment scores $A_k$ stay mostly stable with a minor drop at high $\lambda$ values. We believe this happens as $f$ aims to match the rate-distortion constant, possibly neglecting the underlying representation structure. Second, in the same setup, we modify the equality constraint to be: $|R(\mathcal{Z}) - rb|$ and ablate $r$ (shown in Figure 8 (center)). We observe that both alignment scores and gender accuracy increase with an increase in $r$, which demonstrates the importance of this constraint. Even though $R(\mathcal{Z}|\mathbf{K})$ is maximized, if the overall feature space expands (high $r$), the concept variable can still become distinguishable (high gender accuracy). Third, in Figure 8 (right), we report the MSE scores on the synthetic dataset for varying $\sigma$ (the parameter in the Gaussian kernel). In all setups within Figure 8 (right), we notice the same pattern of increasing MSE ($a$) scores followed by a decrease. We believe this drop happens with higher $\sigma$ values because distances become very small and kernel values are similar. This results in the kernel ignoring the similarity of instances in the concept space.

## E  Broader Impact & Limitations

In this section, we discuss the broader societal impact and limitations of our framework, KRaM.

**Limitations**. While erasing sensitive concept attributes can reduce bias and improve privacy, it may also result in the loss of potentially useful information for the task at hand. This could negatively impact the utility of the model. The definition of what constitutes a sensitive concept attribute can vary greatly depending on the cultural, ethical, and legal context. This work assumes that these sensitive attributes can be clearly defined and agreed upon, which might not always be the case. Therefore, developers using such erasure frameworks should take care of the societal impact before utilizing them in the wild.

**Negative Usage**. KRaM s intended to be used in scenarios where the user is already aware of the concept attribute to be erased. KRaM can only be trained on data where concept labels are annotated either as categorical, continuous, or vector-valued attributes. One potential misuse of KRaM would be to define relevant features for a task (e.g., experience for a job application) as a concept to be erased. In such cases, the classification system may be forced to rely on sensitive demographic information for predictions. It is possible to flag systems in these cases by evaluating the statistical parity when the concept attributes have changed.

In general, we hope that our proposed concept erasure framework, KRaM, would encourage others to develop more robust concept erasure systems that can simultaneously retain a lot of information from the original representations.

