# OpenReview forum: "Robust Concept Erasure via Kernelized Rate-Distortion Maximization"
_NeurIPS.cc/2023/Conference — NeurIPS 2023 poster_

### Official Review · Reviewer_iqZj · 2023-07-06

**Soundness:** 2 fair
**Presentation:** 2 fair
**Contribution:** 2 fair
**Rating:** 5
**Confidence:** 5

**Summary:**

This paper considers the concept erasure task removing an attribute from distributed representations while retaining other information from the original representation space as much as possible. Specifically, it proposes a Kernelized Rate-Distortion Maximizer (KRaM) framework and provides a theoretical analysis for the proposed algorithm. Finally, various experiments are conducted to show the effectiveness of KRaM.

**Strengths:**

The topic considered in this work is important/interesting and the structure of the paper is well-organized.

**Weaknesses:**

1. The contributions of this paper are unclear, and the novelty is also insufficient. From my understanding, the major weakness is that the KRaM is just a simple application of the kernels to the rate-distortion maximizer framework. Meanwhile, the strengths/differences between the proposed KRaM and the current concept erasure methods such as [1,2,3]. The authors should make more discussions in the rebuttal phase.
2. The theoretical results are also limited because it merely applies to the RBF kernel. Thus, it is better to show the theoretical results for other kernels.
3. The experiments are insufficient to support the effectiveness of the proposed KRaM. Because 1) there are a few competitive methods [1,2] that are not considered in this work; 2) the improvements of the KRaM against the current methods are limited. In particular, in Tab.2, compared with RLACE, the performance gap is sharp such as Acc.(y) and A_k.
4. Why do we need to train an extra concept function f? What will happen to the performance if we only use the outputs of a pre-trained model as f?

Ref:

[1] Learning Diverse and Discriminative Representations via the Principle of Maximal Coding Rate Reduction.

[2] Kernelized Concept Erasure.

[3] Probing Classifiers are Unreliable for Concept Removal and Detection.


==================================================

Post-rebuttal:

My concerns are basically solved. So I'd like to raise my score to BA.

**Questions:**

See weakness.

---

> ### Author Rebuttal · Authors · 2023-08-09
>
> Thanks for your review and questions. Please find our responses below.
>
>
>
> (W1)
> We believe there may be a misunderstanding here. And so we would like to clarify things in detail because we firmly believe that our work is not just a simple introduction of kernels to [1]. In fact, **our task and final objectives are distinctly unique and novel** compared to [1]. Let us now specify our reasoning:
>
> * The rate-distortion maximization framework proposed in [1] was for classification tasks, not concept erasure, which relies on the rate-distortion formulation (Equation 1).
> * The original closed formula for rate-distortion (Equation 1 in our paper) was first proposed in [4]. Several works have built on rate-distortion to use it in multiple applications like segmentation [4, 5], classification [1], incremental learning [6],  autoencoders [7], among others.
> * The only work that used rate distortion in fairness is FaRM [8], which we extensively compare in Tables 1 and 2.
> * We have performed new experiments to compare with [2] and report the results on categorical concept erasure below:
>
>
> *DIAL Results*
>
> |Method| Acc. (y) $\uparrow$      | Acc. (a) $\downarrow$      |  DP $\downarrow$ |$A_k$ $\uparrow$ |
> | :---        |    :----:   |:----:   |:----:   |:----:   |
> |Original (w/o concept erasure) |75.5 |87.7 |0.26| 1.0|
> |Kernelized Concept Erasure [2]|75.0 |80.1 | 0.12| 0.70|
> |KRaM (Ours)|72.4 |54.0 | 0.08| 0.74|
>
> *GloVe Results*
>
> |Method| Acc. (a) $\downarrow$ |$A_k$ $\uparrow$ |Rank $\uparrow$|
> | :---        |    :----:   |:----:   | :----:   |
> |Original (w/o concept erasure) |100 |1.0 |300|
> |Kernelized Concept Erasure [2]|63.5 |0.62 | 100|
> |KRaM (Ours)|52.6 |0.65 | 246|
>
> From the above results, KRaM performs significantly better than [2] in erasing concepts achieving far better Acc. ($a$) scores while retaining a significant amount of prior information as seen through the $A_k$ scores. Also, note that the mentioned method can only handle categorical concept erasure while KRaM can perform categorical, continuous, and vector-valued concept erasure.
>
> Regarding [3], we find that this mentioned paper is an analysis work finding several pitfalls of nullspace projection concept erasure approaches. As our proposed method does not rely on a probing classifier during concept erasure we do not share similar vulnerabilities.
>
> [1] Learning Diverse and Discriminative Representations via the Principle of Maximal Coding Rate Reduction.
>
> [2] Kernelized Concept Erasure.
>
> [3] Probing Classifiers are Unreliable for Concept Removal and Detection.
>
> [4] Segmentation of Multivariate Mixed Data via Lossy Data Coding and Compression, Yi Ma, TPAMI 2007.
>
> [5] Unsupervised segmentation of natural images via lossy data compression. Young, 2008
>
> [6] Incremental Learning via Rate Reduction
>
> [7] Closed-Loop Transcription via Convolutional Sparse Coding.
>
> [8] Learning Fair Representations via Rate-Distortion Maximization, TACL 2022.
>
> (W2) The only assumptions that we used in our proof are that $K$ is PSD (holds for all kernels) and $d(x, x)=0$, $K(0)=1$. The condition $K(0)=1$ holds for a lot of kernels. We will restate our lemmas to include these exact conditions.
>
> Moreover, if we consider the general case where $K(0)=c$, $c$ being a constant, we modify the bounds as follows:
>
> $c^nR(Z) \leq R(Z|K) \leq \frac{n}{2}\log_2 (1+\frac{cd}{n\epsilon^2})$
>
> Proof sketch: The above can be proved by simply replacing $b_{ii}=c$ in line 537 and $\lambda_i(ZZ^T \odot K) = cn/n=c$.
>
>
> (W3) *Concern 1*: We have added results for the mentioned baselines and reported the results above as a response to your first question.
>
> *Concern 2*: We would like to add more context to help the reader navigate the results better.
>
> The primary goal of concept erasure is to delete a specified concept even at the cost of removing additional information. Unlike adversarial learning, in this application, we are not trying to achieve a balance between fairness and target tasks. However, we need to ensure that the network doesn’t produce random representations and lose all information in the process. We evaluate this using $A_k$ and other downstream tasks.
>
> RLACE achieves better $A_k$ and Acc. ($y$) because it cannot remove the desired concept properly. If you observe the Acc. ($a$) (in Table 2) of RLACE it is significantly higher than KRaM. This shows that it is unable to delete concepts robustly, which is the primary objective. This phenomenon is consistently observed in Table 2 where RLACE reduces gender accuracy by only 5% (compared to 47.2% KRaM) and DP improvement on DIAL is 0.08 by RLACE compared to (0.18 by KRaM).
>
> In general, slightly lower Acc. ($a$) and $A_k$ obtained by KRaM indicates that other information often gets compromised in order to delete the concept robustly.
>
> (W4) We performed experiments with outputs from a pre-trained model. Specifically, we use GPT-3.5 representations for vector-valued concept erasure reported in Figure 3.
>
> The motivation behind concept erasure stems from the application where a data producer is distributing representations to different customers and erasing specific concepts based on the customer’s application. In this case, it is beneficial to use a separate function $f$ for each customer so that the underlying foundation model can be reused. In theory, the user can treat the whole foundation model as $f$ and train it for concept erasure. But the model would not be useful in future applications that need the deleted or related concepts.

---

> > ### Author Response · Authors · 2023-08-14
> >
> > Thanks for taking the time to review our responses. Please let us know if you have any questions or would like any further clarifications.

---

### Official Review · Reviewer_x4Ej · 2023-07-06

**Soundness:** 3 good
**Presentation:** 2 fair
**Contribution:** 3 good
**Rating:** 6
**Confidence:** 4

**Summary:**

The KRaM framework is proposed in this paper to perform concept erasure. The approach aims to differentiate similar concept labels in the learned representation space while preserving other information. The paper demonstrates the effectiveness of KRaM in erasing various concepts across different forms of representation (i.e., categorical, continuous, and vector-value variables) through theoretical analysis and empirical setups. Additionally, the paper proposes a clear measurement to evaluate the alignment of representations before and after concept erasure.

**Strengths:**

This paper presents several strengths, including a clear and well-written introduction that guides the reader to understand the problem and the motivations behind it, as well as providing helpful background study. They provide clear setups to study and offer results empirically and theoretically. This paper proposes a new approach to address the limitation of prior works, which did not evaluate the amount of information preserved in the representation. The proposed method uses a clever and heuristic idea to measure alignment by comparing overlapping neighborhoods of the original representation.

**Weaknesses:**

Here are some potential improvements to the original content:

1. In line 273, it is stated that a more effective method of erasing concepts would result in low accuracy and high mean squared error (MSE) for the deleted concept. This statement may seem counterintuitive, since KRaM claims to retain some original information. However, as shown in Figure 3, the performance of downstream task drops relatively significantly when concepts such as Christian, Jewish, and Muslim are erased. The relationship between utility and representation preservation needs more explanation.

2. The author has presented a heuristic approach to determine the amount of information that is retained in a representation. However, it is unclear how this approach can guarantee that the representation is altered according to the given concept, particularly in the continuous form of representation.

**Questions:**

Is it possible to apply KRaM to image-based datasets? Doing so could potentially enhance the proposed method's ability to generalize across various modalities.


**Limitations:**

The paper presents a novel framework that should potentially be applied to various modalities. Finding and understanding the exact information which is altered by concept erasure is a future work and has positive societal impact to the field.

---

> ### Author Rebuttal · Authors · 2023-08-09
>
> Thank you for your review and questions. Please find our responses below.
>
> Weaknesses:
>
> (W1) We would like to clarify the claim made in the paper.
>
> We emphasize that the purpose of concept erasure is to make it difficult to predict the deleted concept from the representation set. And indeed, our empirical results show that prediction of deleted concepts is difficult post-concept erasure. Poor prediction performance means that any classifier trying to predict the deleted concept would obtain low accuracy (for categorical variables) and high MSE (for continuous variables).
>
> In Figure 3, we **do not report the downstream task** performance but performance of detecting the deleted concept. The downstream performance is reported in Line 300. We observe that there is minimal impact on the toxicity classification performance: pre-concept erasure: 93.2% $\rightarrow$ post-KRaM 92.1%
>
> In this setup reported in Figure 3, we erase vector-valued concepts where the vector has dimensions [buddhist, christian, hindu, jewish, muslim, others]. After deleting these vector concepts, we probe the modified representations for each of the religion attributes individually (reported in Figure 3(a)) and compute the fairness in the downstream task using GDP values, Figure 3(b) (where lower GDP means indicates a fairer predictor). In Figure 3(a), we observe that the MSE of predictors trying to predict religion increases significantly after KRaM. This implies that the modified representations contain much less information about religion, which is expected.
>
> (W2) We would like to clarify the intuition behind the proposed approach.
>
> The approach **does not** measure the information content at the level of individual representations. Instead, the alignment score is used to compare the information content between two sets of representations. The motivation behind this approach comes from the fact that the nearest neighbour structure in the representation space contains useful information for downstream tasks. Consider the case of classification using a kNN classifier, where a test sample is assigned the majority label of its nearest neighbours. Now, if the nearest neighbor structure is destroyed it won’t be possible to predict the right labels and therefore the downstream task information is lost. Note that the alignment score is used to measure overall information alignment and does not account for the concept being erased.
>
> In all of our experiments, we evaluate the representation quality of KRaM using multiple different methods apart from $A_k$ scores like downstream tasks, rank, and word embedding similarity as reported in Tables 1, 2 and Figure 4.
>
> Questions:
>
> (Q1) Yes, it is possible to perform concept erasure using KRaM on image-based datasets. We report the results of concept erasure on two different datasets below
>
>
> *CelebA*:
>
> Setting:
> * Dataset: We select a subset of CelebA, which contains images of people’s faces
> * Target binary label: attractiveness \in [attractive, not attractiveness]
> * Concept to be erased: whether the face [has makeup, doesn’t have makeup]
>
>
> Observations:
> *  Prior to the concept-erasure makeup factor (concept to be deleted) could be predicted with 85.7% accuracy
> * Post KRaM the prediction accuracy decreased to 69.6%
> * The demographic parity of the predictions also improved from 0.94 to 0.54. This shows the effectiveness of KRaM in removing the makeup concept.
>
>
> *Colored MNIST*:
>
> Setting:
>
> * Dataset: We follow the setup of [1] to create a biased version of the MNIST dataset.
> * Labels: The background color of the digits is made to be well correlated with the digit in the training set. But no such correlation exists in the test set. We use 0.8 correlated between the background color and digit label.
> * Concept to be erased: Background color is the concept to be erased.
>
> Observations:
>
> * Naively training a classifier on the images will perform poorly on the test set as the classifier can easily overfit on the background color. The digit information could be predicted with an accuracy of 79.9%.
> * Post-concept erasure we observe that the classifier’s performance improves from 79.9% to 87.1%. This shows the effectiveness of KRaM to remove background information and help the classifier generalize better.
>
>
> [1] Learning De-biased Representations with Biased Representations, Bahng et al. ICML 2020.

---

> > ### Author Response · Authors · 2023-08-14
> >
> > Thanks for taking the time to review our responses. Please let us know if you have any questions or would like any further clarifications.

---

> > > ### Comment · Reviewer_x4Ej · 2023-08-15
> > > **Thanks for your clarifications!!**
> > >
> > > Dear Authors,
> > >
> > > Thanks for your response and detailed clarifications. About the W1, I have read the paper again and correct my understanding to Ln 273. And thanks for offering additional image-based experiments, which looks great and promising. I believe this can be included in the paper and enhance paper's contribution. I raised my score by 2 and support this paper to be accepted. Thanks!

---

### Official Review · Reviewer_GgzL · 2023-07-06

**Soundness:** 3 good
**Presentation:** 3 good
**Contribution:** 3 good
**Rating:** 7
**Confidence:** 3

**Summary:**

The paper proposes a novel method to erase concepts from a distributed/latent data representation. The method is based on a kernelized rate-distortion function that can be maximized to achieve concept erasure. The kernel encodes the "concept similarity" between two samples, and it is therefore supervised. That is, it needs a concept label associated with the samples.

From a set of experiments, the authors show that in general they manage to achieve a more effective concept erasure without loss of performance in downstream tasks.

**Strengths:**

- (Significance) Concept erasure is a timely topic for AI safety, and an interesting topic per se in the broader interpretability field.
- (Novelty) Kernel-trick, coupled with rate-distortion theory seems like a reasonable novel contribution.
- (Soundness) Experiments seem convincing to support authors claims

**Weaknesses:**

- (Soundness) Assumptions and drawbacks should be further clarified. See section below.
- (Theoretical results) I am unsure how useful is the proven lemma in the overall narrative of the paper. The only mentioned conclusion is the one in line L.202. However, this conclusion can be reached by simply looking at equation 2. And actually the conclusion is wrong if we were to only look at the inequality: since R(Z) is the *lower* bound, increasing R(Z/K) does not necessarily imply an increase in R(Z).

**Questions:**

- Gaussianity assumption: have the authors tested with different formulations of the rate distortion function? would this be feasible?
- Determinant scalability: the determinant seems to badly scale with the number of samples in Z. Could the authors comment on the computational bottleneck of their method, and possible ways to solve it?
- Additional assumptions: I believe there should also be an assumption about the concepts being uncorrelated. I would imagine that if different concepts are correlated in the dataset, then the authors method could accidentally erase more concept, e.g. using one of the experiments, erasing religion could potentially erase also ethnicity information. Could the authors comment on this, potentially also from the perspective of societal impacts.
- As a sanity check, have the authors tested their method on the bottleneck layer of a concept bottleneck neural network [Koh, 2020]? I would imagine that if the method worked properly, then the authors could clearly see that one of the dimension in the bottleneck layer would essentially become constant.

**Limitations:**

Only few limitations are discussed from a techincal perspective. Societal impacts are also very briefly mentioned.

---

> ### Author Rebuttal · Authors · 2023-08-09
>
> Thank you for your review and questions. Please find our responses below.
>
> Questions:
>
> (Q1) We are unfamiliar with and did not get to try alternative formulations of the rate-distortion function. However, even in the non-parametric setting where the representations are available as finite samples, the rate-distortion formula (Equation 1 in our paper) provides a good upper bound. This has been shown in Appendix A of this paper [1]. We had mentioned it briefly in Lines 131-132, but will add more details in the paper.
>
> [1] Segmentation of Multivariate Mixed Data via Lossy Data Coding and Compression, Yi Ma, TPAMI 2007.
>
>
> (Q2) Indeed, the determinant scales badly with the number of samples in $Z$. In practice, the network is trained using mini-batches where the number of samples is significantly smaller compared to the dataset size. Even with relatively small mini-batch sizes  we are able to achieve good performance with KRaM. The batch size provides one simple way to mitigate computation bottlenecks based on the available resources. In practice, we found our method to be quite fast, for a batch size of 200, dataset size ~10K, representation size 300, 100 epochs of training completed within 1 minute on an Nvidia Quadro RTX-5000 GPU.
>
> Other works that focused on variational formulations of a different rate-distortion-based objective function in Proposition 3.1 in [2] for efficiently computing the rate distortion. We view this as orthogonal to the contributions of KRaM.
>
> [2] Efficient Maximal Coding Rate Reduction by Variational Forms, Christina Baek, CVPR 2022.
>
> (Q3) Yes, your observation is correct. If the user is erasing a concept label using KRaM (or other concept erasure frameworks) then other correlated concepts can also be erased. In the extreme case, you can think about retrieving a concept that is 100% correlated with the erased concept, which won’t be possible. In general, the correlation between different concepts or latent factors depends on the modeling techniques used to produce the data representations and it is difficult to provide guarantees.
>
> We briefly talk about this in the broader impact section in Appendix E and will add more details in the paper.
>
> (Q4) Thanks for your suggestion.
>
> Due to time constraints, we experimented with a simpler setup involving concept bottleneck neural networks, which is a step towards the experiment you suggest.
>
> *Setting*:
> *  Trained a model (inception v3) to predict concepts given an image.
> * Performed concept erasure using KRaM on the representations from this model.
> * Post-concept erasure the representations were used to predict the target task (with 200 labels) using a separate classifier.
> * Since *this classifier has access only to the concepts and not the original image* performing concept erasure should make it incapable of predicting the target class.
>
> We perform vector-valued concept erasure to delete a subset of the concepts (there are a total of 112 attributes). The results are reported below:
>
> |% Concepts Erased| Acc. (y) |
> | :---        |    :----:   |
> |0% |69.5 |
> |12.5% |42.7 |
> |25% |37.0 |
> |50% |35.4 |
> |100% |0.08 |
>
> We observe that as more concepts are erased the target task performance decreases. We observe that when 100% of the concepts are erased the representation cannot be used to predict the target task, which shows the effectiveness of KRaM. We draw this conclusion because the representations include only features corresponding to their concepts and not the underlying image based on the way the model was trained.
>
>
> Weaknesses:
>
> (W2) Thanks for pointing this out. Your observation is correct and the statement is incorrect. However, we did observe this empirically where maximizing $R(Z|K)$ also results in an increase in $R(Z)$ (Section 5.2 in Figure 6(a)). We will modify the statement in the paper.
>
> Apart from this, we find that the bounds of $R(Z|K)$ can be used to find the overall bound for the objective in Equation 4.
>
> $[-\lambda b, \max((1+\lambda)U-\lambda b, (1-\lambda)U+\lambda b)]$
>
> where $U = \frac{n}{2} \log_2 (1 + d/n \epsilon^2)$ and assuming $\lambda \in [0, 1]$ is the upper bound derived in Lemma 1.

---

> > ### Author Response · Authors · 2023-08-14
> >
> > Thanks for taking the time to review our responses. Please let us know if you have any questions or would like any further clarifications.

---

> > ### Comment · Reviewer_GgzL · 2023-08-18
> > **Thanks for the rebuttal**
> >
> > I thank the authors for the insightful rebuttal. Most of my concerns were addressed, however I am unsure about the additional experiment provided.
> > The table provided does not seem to directly address my concern since, if I am understanding correctly, the additional experiment is essentially the same as the ones already provided in the submission (E.g. figure 3).
> >
> > What I would have like to see was more something similar to Figure 5, but for the very specific case of concept bottleneck models, or similar models where we know that the concepts are aligned to the canonical base, and therefore I find it to be a very nice sanity check to see that only the dimension of the erased concept is affected. I would find this to be better evidence than Figure 5 since an additional step (UMAP) is needed for the visualization, which makes it more difficult to fully give credit to Kram for the concept erasure.
> >
> > Actually thinking about it, your method does not need to be applied to a latent representation from a neural network, right? Theoretically speaking, for any tabular dataset, the original feature of the input data could be regarded as concepts. In other words, if you were to apply your method to, say, the Iris dataset to delete the "concept" "petal width", we should see an effect only for that feature, right? Would you be able to run such an experiment? Should be quick I believe.

---

> > > ### Author Response · Authors · 2023-08-18
> > >
> > > Thank you very much for your reply and your continued clarifications and suggestions.
> > >
> > > Motivated by your suggestion, we have conducted an additional experiment. We will provide the details for this experiment following our explanation of how concept erasure handles deleting an axis-aligned concept.
> > >
> > > *Note about removing axis-aligned attributes* We wanted to clarify that having an axis-aligned feature be zero does not necessarily mean that the concept has been deleted. For example, it may be still possible to infer the deleted concept using a linear combination (or a non-linear function) of the other features. Rather it is the case that the dimensionality of the subspace of the data post-concept deletion will be reduced. In addition to that, the predictive power of classifiers trying to predict the deleted concept should reduce significantly.
> > >
> > > To illustrate this further, we design an experiment, following your suggestion, with axis-aligned concepts [NB 1].
> > >
> > > *Experiment*: IRIS has 4-dimensional data; instead we consider a simpler, easier to visualize setting.
> > >
> > > * Generate a synthetic 2D dataset involving two Gaussians centered at (0, 2) and (0, -2) respectively, and identity covariance.
> > > * Consider the $y$-axis as the continuous concept that will be deleted [NB 2]
> > >
> > > The *desired outcome* of deleting an axis-aligned concept in 2D data is that points should lie on a 1D line and the KRaM output should be least correlated with the axis concept that was deleted. That is a reduction in dimension such that most of the data variation should be constrained along a single dimension for this case.
> > >
> > > *Results*: We do observe that most points lie on a 1D line and their variation is constrained along a single dimension. [NB 3]
> > >
> > > *How do we know the correct axis information has been deleted?* We color-code the post-processed representations along with the original $y$-axis values and find no correlation among the points. We further tried predicting the concept from the original data and the KRaM output and found the MSE (for concept) had increased from 0.00025 (original) to 0.77 (KRaM) further showcasing the effectiveness of KRaM.
> > >
> > > Since we are unable to show plots we wanted to give some numbers to showcase that KRaM output indeed lies on a line (one dimension less than the input). For this, we looked at the eigenvalues of both the original data and the concept deleted (by KRaM) output. The fraction of eigenvalue masses in the original data [65%, 35%] and in the KRaM concept deleted output the fraction of masses is [99.998%, 0.002%].
> > >
> > > Of course, one can perform rotation of the resultant points to align them exactly with the $y$-axis, which would result, as you suggested, in the $y$-axis becoming zero.
> > >
> > > [NB 1]  We are unable to share the plots because we cannot edit the response PDF during the rebuttal. NeurIPS also doesn’t allow sharing any anonymous links. We will include the results in the paper.
> > >
> > > [NB 2] We produced a plot in which the points are color-coded according to the $y$-axis value, i.e., the color attribute is the one that will be deleted.
> > >
> > > [NB 3] We produce a figure that shows each point in the dataset post-concept deletion. Indeed, the result is one line per Gaussian such that the color of points along the line is entirely mixed up without any clear or generalizable pattern.

---

> > > > ### Comment · Reviewer_GgzL · 2023-08-18
> > > > **Great!**
> > > >
> > > > Thanks for the additional experiment!
> > > >
> > > > I am leaning towards raising the score to 7 already, but if you allow me to be a lil bit pickier...
> > > >
> > > > Your result is in support of your method working under gaussianity assumption. For the sake of "real-world" applicability, could you apply it still to the IRIS dataset just by discarding one of the features (since we are not caring about classification performance)?
> > > >
> > > > Does the method still perform well? If not, is it solely because of your formulation? Or could there be other reasons?

---

> > > > > ### Author Response · Authors · 2023-08-19
> > > > > **Thanks! Results on IRIS**
> > > > >
> > > > > We appreciate your positivity towards the paper and leaning towards increasing your score.
> > > > >
> > > > > Here are the results for the experiment you suggested on IRIS (erasing the petal length feature). We divided the data randomly into 80%/20% train/test split and report the results on the test set.
> > > > >
> > > > > We see that the erasure leads to an output that lies in fewer dimensions than the input. The minimum eigenvalue mass changes from 1.58% [80.59, 14.91,  2.90,  1.58] to 1.5e-6% [81.79, 17.79, 0.42, 1.5e-06]. While the MSE increases from 0.0033 (original data) -> 0.0114 (KRaM output).
> > > > >
> > > > > Thank you very much for being so engaged in this discussion! We really appreciate it!
> > > > >
> > > > > We will add these results to the paper.

---

### Official Review · Reviewer_yWDJ · 2023-07-27

**Soundness:** 3 good
**Presentation:** 4 excellent
**Contribution:** 3 good
**Rating:** 7
**Confidence:** 3

**Summary:**

This paper studies concept erasure, a task proposed by prior work [1,2] which aims to remove information from a chosen attribute while retaining other information from the original input as much as possible. This paper achieves this by maximizing the kernelized rate-distortion maximizer (KRaM), an objective from information theory measuring how well lossy compression methods obtain the original information—specifically, maximizing the objective results in increased dissimilarity between instance pairs with similar concept labels, leading to the distances in the learned representation unrelated to the concept labels, achieving concept erasure. To preserve the information from the original input, a constraint is proposed in this paper by maintaining the rate-distortion function to a constant size using a Lagrangian function. The authors propose to use k-nearest neighbors to measure the alignment: how much the learned representation preserves the information from the original input after concept erasure. Bounds for the objective and expectation of the alignment are provided. Finally, the authors provide empirical results to demonstrate that the proposed method can erase concepts (categorical, continuous, and vector-valued) in the DIAL, UCI Crimes, and Jigsaw toxicity datasets.


[1] Ravfogel, Shauli, et al. "Kernelized Concept Erasure." arXiv preprint arXiv:2201.12191 (2022).

[2] Ravfogel, Shauli, et al. "Linear adversarial concept erasure." International Conference on Machine Learning. PMLR, 2022.


**Strengths:**

Originality: The research problem is already defined and studied, but the method of using kernelized rate-distortion function for concept erasure is new.

Quality: The quality of this submission is relatively high. The related publications are cited and discussed. The empirical analysis is comprehensive, and the theoretical results are also provided. Notably, the method can erase concepts from categorical attributes and continuous or vector-based concepts (e.g., word vectors). A synthetic experiment verifies the proposed measuring alignment and outperforms other alignment measurements. The empirical results on vector-valued concept erasure are convincing.

Clarity: The paper is well-written. Key terms, such as concepts, the rate-distortion function, and the alignment score measurement, are clearly defined.

Significance: The authors make a great effort to use the proposed KraM on distributed word vector representations such as GPT-3.5 to erase concepts, making this work very relevant and significant to the recent trends of large language models and valuable for many real-world tasks requiring removing certain concepts or attributes.


**Weaknesses:**

> Instead, objectives for concept erasure use no supervision (apart from the labels of the concept to remove). This makes using adversarial learning or mutual information estimation methods for concept erasure difficult.

Unfortunately, this may not be the case. Although the specific problem of concept erasure is not studied in mutual information estimation methods, works such as [1-3] remove information from a particular attribute without any downstream task known using mutual information estimation or invariant learning. It would be helpful for the reviewer to give final recommendations if the authors could further compare these works.

Also, the proposed method has similar or worse results than baselines, such as some entries in Table 1.

[3] Song, Jiaming, et al. "Learning controllable fair representations." The 22nd International Conference on Artificial Intelligence and Statistics. PMLR, 2019.

[4] Tsai, Yao-Hung Hubert, et al. "Conditional contrastive learning: Removing undesirable information in self-supervised representations." arXiv e-prints (2021): arXiv-2106.

[5] Moyer, Daniel, et al. "Invariant representations without adversarial training." Advances in Neural Information Processing Systems 31 (2018).

**Questions:**

In Figure 4, the accuracy and alignment score trend differs from Section 4, where the authors claim that the alignment score is well correlated with the downstream performance. The reviewer is unsure about this and gently asks the authors to clarify.

**Limitations:**

The authors have discussed the limitations of the work as well as future directions.

---

> ### Author Rebuttal · Authors · 2023-08-09
>
> Thank you for your detailed review. Please find our responses below:
>
> Weaknesses:
>
> (W1) Thank you for pointing out these works. We will remove the specific claim and cite these works. We have compared with the suggested works and the results on categorical concept erasure are reported below:
>
> *DIAL Dataset Results*
>
> |Method| Acc. (y) $\uparrow$      | Acc. (a) $\downarrow$      |  DP $\downarrow$ |$A_k$ $\uparrow$ |
> | :---        |    :----:   |:----:   |:----:   |:----:   |
> |Original (w/o concept erasure) |75.5 |87.7 |0.26| 1.0
> |MIFR (Song et al. 2019, [3])|75.4|68.7|0.21|0.88|
> |CCL (Tsai et al. 2019, [4])|50.7|52.6|0.01|0.60|
> |ICVAE (Moyer et al. 2018, [5])|66.5|53.3|0.10|0.55|
> |KRaM (Ours)|72.4 |54.0 | 0.08| 0.74|
>
> *GloVe Results*
>
> |Method| Acc. (a) $\downarrow$ |$A_k$ $\uparrow$ |
> | :---        |    :----:   |:----:   |
> |Original (w/o concept erasure) |100 |1.0 |
> |MIFR (Song et al. 2019, [3])|68.3|0.58|
> |CCL (Tsai et al. 2021, [4])|56.9|0.56|
> |ICVAE (Moyer et al. 2018, [5])|51.5|0.50|
> |KRaM (Ours)|52.6 |0.65 |
>
> Note the following limitations of these methods:
> * ICVAE requires access to concept labels for test instances as well, a limitation compared to KRaM and the other methods listed.
> * Moreover, CCL, ICVAE, MIFR can function only for categorical attributes.
>
> Here are our conclusions from the above results:
> * MIFR is unable to erase concepts robustly (see Acc. (a) and DP scores).
> * ICVAE & CCL is able to delete concepts but at the significant cost of deleting a lot of original information.
> * Loss of information from the original space (low $A_k$) using ICVAE and CCL is quite similar to other MI methods reported in Figure 4 of our paper (about ~0.9-0.29 points behind KRaM in $A_k$).
> *  KRaM is able to get similar concept erasure performance while achieving much higher $A_k$ and Acc. ($y$) scores in both setups.
>
> Questions:
>
> (Q1) Thanks for the great question.
>
> The experiment was conducted on the GloVe dataset where male and female-biased words form distinct clusters.
>
> We observe a unique scenario for the CLUB method. The gender prediction accuracy is high and the $A_k$ is low. This implies that the clusters (related to different genders) may be retained but the nearest neighbour structure within the clusters is perturbed significantly.
>
> Probing for the downstream task (gender accuracy) may give you the impression that information from the original space is retained, while $A_k$ provides a more fine grained view contradicting such incorrect conclusion.
>
> We find that $A_k$ values are relatively well correlated with the WS-353 scores.
> WS-353 is a good measure of the information retained in the representation space after erasure . Computing the WS-353 however requires additional annotation that may not be feasible for representation sets other than word embedding.

---

> > ### Author Response · Authors · 2023-08-14
> >
> > Thanks for taking the time to review our responses. Please let us know if you have any questions or would like any further clarifications.

---

> > ### Comment · Reviewer_yWDJ · 2023-08-16
> > **Thank you**
> >
> > This reviewer's question and concern have been addressed. Therefore, the reviewer raises the rating.

---

### Author Rebuttal · Authors · 2023-08-09

We thank the reviewers for their valuable feedback and suggestions. We believe that we have carefully considered all the comments raised by the reviewers. We have conducted several additional experiments and theoretical analysis as summarized below:

# New Experiments & Results

**New Baselines**:

  * Mutual Information-based bounds for categorical concept erasure
    * Methods: MIFR, CCL, ICVAE
    * Datasets: DIAL & GloVe
    * Observation: KRaM is able to remove concepts robustly while retaining significant amount of prior information compared to baselines.
    * Report to Reviewer yWDJ [here](https://openreview.net/forum?id=I6aOjhpcNQ&noteId=Qows1Jde4O).
* Analysis using concept bottleneck neural networks
  * Model trained & concept erasure experiment on CUB dataset
  * Observation: As more concepts are erased it becomes difficult to predict target labels from the concept bottleneck layer (which is the desired outcome).
  * Report to Reviewer GgzL [here](https://openreview.net/forum?id=I6aOjhpcNQ&noteId=o1xDLck1hN).
* Image Domain experiments:
  * Evaluated KRaM on two datasets: CelebA & Colored MNIST
  * Observation: KRaM is able to remove concepts robustly leading to improved DP and generalization respectively.
  * Report to Reviewer x4Ej [here](https://openreview.net/forum?id=I6aOjhpcNQ&noteId=u2Nfl0IqJy).
* We have compared KRaM with the kernel concept erasure method on GloVe and DIAL datasets. The results are reported in the response to Reviewer iqZj.
* Comparison to Kernelized Concept Erasure:
  * Dataset GloVe and DIAL
  * KRaM achieves better concept erasure and information retention scores than this baseline.
  * Report to Reviewer iqZj [here](https://openreview.net/forum?id=I6aOjhpcNQ&noteId=YouK0TQGUL).

# Theoretical Analysis

* We have clarified the statements related to Lemma 1. We also added results about how these bounds can be utilized to find the bounds of the overall objective (response to Reviewer GgzL [here](https://openreview.net/forum?id=I6aOjhpcNQ&noteId=o1xDLck1hN)).
* We have clarified the exact conditions on the kernel functions required for lemma 1 to hold. We have also added the results for a general class of kernels (response to Reviewer iqZj [here](https://openreview.net/forum?id=I6aOjhpcNQ&noteId=YouK0TQGUL)).

---

### Decision · Program_Chairs · 2023-09-21

**Decision:**

Accept (poster)

**Comment:**

This paper studies a relatively new area in the broad area of machine unlearning. It specifically studies "concept" erasure, where the goal is to take a set of representations and remove the influence of a particular concept while leaving other information unchanged. The key idea is to adopt a kernelized rate-distortion framework to perform erasure. This is not a new idea, but the authors generalize it to handle a lot of possible notions of "concept".

Reviews are largely quite positive, noting good results trading off erasure ability vs. maintaining information. This is a solid contribution to an important developing area.